# BENFEP, a quantitative database of BENthic Foraminifera from surface sediments of the Eastern Pacific

Paula Diz[1], Víctor Gónzalez-Guitián[1,2], Rita González-Villanueva[1], Aida Ovejero[3], Iván Hernández-Almeida[4]

[1]Centro de Investigación Mariña, Universidade de Vigo, XM1, 36310 Vigo, España
[2]Instituto de Investigaciones Marinas, CSIC, Eduardo Cabello 6, 36208 Vigo, España
[3]Cátedra UNESCO en Desarrollo Litoral Sostenible, Universidade de Vigo, 36310 Vigo, España
[4] Geological Institute, ETH Zürich, 8092 Zürich, Switzerland

*Correspondence to*: Paula Diz (pauladiz@uvigo.es)

**Abstract.** Benthic foraminifera are important components of the ocean benthos and play a major role in ocean biogeochemistry and ecosystems functioning. Generating ecological baselines for ocean monitoring or biogeographical distributions requires a reference dataset of recent census data. Besides, the information from their modern biogeography can be used to interpret past environmental changes on the sea-floor. In this study, we provide the first comprehensive quantitative BENthic Foraminifera database from surface sediments of the Eastern Pacific (BENFEP). Through the collation of archival quantitative data of species abundances and its homogenization according to the most recent taxonomic standards, we are able to provide a database with 3077 sediment samples, corresponding to 2509 georeferenced stations of wide geographical coverage (60ºN and 54ºS) and water depths (0-7280 m). The quantitative data includes living, dead, and living plus dead assemblages obtained from 50 published and unpublished documents. As well as describing the data collection and subsequent harmonization steps, we provide summarized information of metadata, examples of species distribution, potential applications of the database and recommendations for data archiving and publication of benthic foraminiferal data. The database is enriched with meaningful metadata for accessible data management and exploration with R software and geographical information systems. The first version of the database (BENFEP_v1) is provided in short and long format and it will be upgraded with new entries and when changes are needed to accommodate taxonomic revisions.

## 1 Introduction

The Eastern Pacific extends from the tidewater glaciers at Alaska to the fjords of Chile, encompassing a habitat that integrates eight Large Marine Ecosystems (Sherman, 1991) covering 10.7 million of ha (Fig. 1). Tropical and subtropical latitudes harbour exceptional levels of pelagic and benthic biodiversity and the presence of endemic species at macro and microorganism's levels (e.g., Davies et al., 2017; Gooday et al., 2021). Several areas of Eastern Pacific Ocean are at severe risk of species loss (Finnegan et al., 2015; Yasuhara et al., 2020, UNESCO, 2022) and consequently, some of them have been categorized as marine protected areas (Enright et al., 2021).

The Eastern Pacific is influenced by ocean-atmosphere natural climate variability modes at decadal-to-multidecadal (e.g., El Niño-Southern Oscillation, the Pacific Decadal Oscillation, the North Pacific Gyre Oscillation; Stuecker, 2018), millennial (Pisias et al., 2001) and glacial-interglacial (e.g., Walczak et al., 2020) timescales. These processes resulted in changes in temperature (Liu and Herbert, 2004), salinity (Praetorius et al., 2020), and productivity (Costa et al., 2017) in the surface ocean and oxygen concentrations in the bottom waters (Cannariato and Kennett, 1999). In an historical context, the increase in ocean temperatures and the expansion of the already existing extensive oxygen minimum zone (found about 100 to 900 m water depth, Karstensen et al., 2000) are the major threats to shallow and deep-water benthic ecosystems from the Eastern Pacific (Sweetman et al., 2017; Breitburg et al., 2018; Yasuhara et al., 2019). These attributes make the Eastern Pacific an area of

interest for assessing the past, present, and future of the marine ecosystem status and its response to expected environmental changes (e.g., Calderon-Aguilera et al., 2022).

The Ocean Decade Implementation Plan (2021-2030) (https://www.oceandecade.org/), promoted by the United Nations, establishes several priority objectives for ocean sustainable development and conservation, which include a more profound

understanding of benthic ocean ecosystem functioning and a better assessment of the vulnerability of coastal and deep ocean areas to ongoing impacts of anthropogenic activities and climate change. Attaining such targets might be challenged by the scarcity and unevenness of recent benthic organisms' census data that might function as suitable natural baselines (Yasuhara et al., 2012; Kidwell, 2015; Borja et al., 2020). Benthic foraminifera, microscopically-sized and shelled organisms (generally ranging from 63 μm to 1000 μm, Murray, 2006) are major components of the marine benthos. Those whose shells are

composed of calcium carbonate have the potential of being preserved in the marine sediments, providing an ideal natural archive for recording past seafloor conditions.

Benthic foraminifera have been used for decades as past environmental indicators (Jorissen et al., 2007) and, more recently, in environmental monitoring (Alve et al., 2016; Jorissen et al., 2018). For example, in the Eastern Pacific, benthic foraminifera

were used as proxies for changes in productivity (e.g., Patarroyo and Martinez, 2015; Diz et al., 2018, Tapia et al, 2021) and intermediate and deep-water oxygenation (e.g., Cannariato and Kennett, 1999; Tetard et al., 2017; Sharon et al., 2020). The proxy value of benthic foraminifera as palaeoenvironmental or biomonitoring tools could be hampered if the full scope of current biodiversity patterns, spatial distributions, and species-environmental relations are not fully known or grounded on a limited number of observations (e.g., Jorissen et al., 2007). A synthesis effort of recent benthic foraminiferal quantitative

occurrences would definitively lead to attain a more complete picture of biogeographical distributions and relationships between environmental parameters and species composition, rendering interpretation of the fossil record more meaningful.

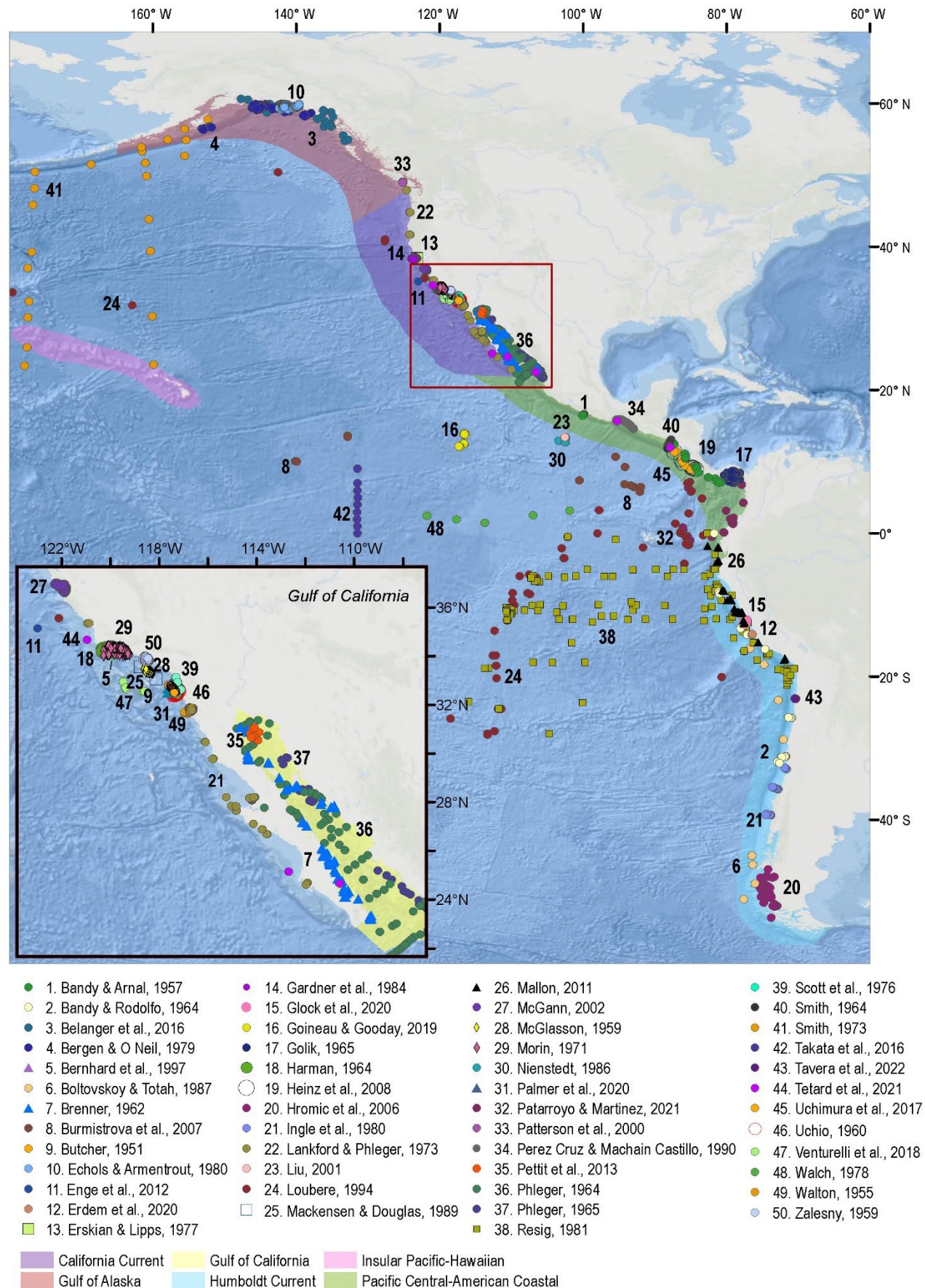

**Figure 1:** Spatial distribution of the samples comprising the BENFEP_v1 database. The numbers refer to each author´s dataset (see Table A1 for additional information). Colour-shaded areas represent the Large Marine Ecosystems of the Eastern Pacific. The map was made using ArcGIS software version 10.8.2 The global relief model integrates land topography and ocean bathymetry (Sources: Esri, Garmin, GEBCO, NOAA NGDC, and other contributions).

The data synthesis of marine microfossils from surface sediments has been a valuable resource among the palaeoceanographic community. They are generally used for constructing modern analogues to interpret the fossil record and, more recently, to evaluate the biodiversity response to ongoing climate change (e.g., Jonkers et al., 2019; Yasuhara et al., 2020). However, existing compilations of marine microfossils covering large ocean swathes mainly integrate census data of planktonic organisms dwelling in the first hundred meters of the water column, such as planktic foraminifera (Siccha and Kucera, 2017), dinoflagellates (Marret et al., 2020), radiolarian (Boltovskoy et al., 2010; Hernández-Almeida et al., 2020), diatoms (Leblanc et al., 2012) or coccolithophores (Krumhardt et al., 2017). Public databases focused on quantitative surface distribution of benthic microfossils are being developed for ostracods (e.g., Cronin et al., 2021, see also review by Huang et al., 2022). Existing quantitative benthic foraminifera datasets from surface sediments including a relatively large number of stations (<300) are restricted to specific ocean sectors, size fractions, or test nature. Examples of these publicly available benthic foraminifera databases are those developed for the Norwegian continental shelf (Sejrup et al., 2004), which includes 298 stations and contains only calcareous foraminifera; the Indian Ocean (De and Gupta, 2010), with 131 core-top samples; or the central Arctic Ocean (Wollenburg and Kuhnt, 2000), with 90 stations. In the East Pacific, the science community performed sporadic research efforts to attain an overview of the quantitative distributions of benthic fauna (e.g., Lankford and Phleger, 1973, n=102; Resig, 1981, n=121; Loubere,1994, n=66, n indicates the number of samples with quantitative data). However, the large area to cover and the economic and time-wise efforts required to sample a significant portion of the sea-floor sediments of the entire Eastern Pacific have prevented the construction of a large and consistent database of benthic fauna for this region.

In this paper, we present BENFEP, a quantitative database of BENthic Foraminifera from surface sediments of the Eastern Pacific. The first version (BENFEP_v1)  contains a rich collection of metadata (e.g., research vessel, sampling devices, processing methods, etc) and quantitative data  (presented in percentage, counts, and densities) of harmonized benthic foraminiferal taxa obtained from more than 3000 samples of living, dead, and living plus dead assemblages gathered from published and unpublished studies. Here, we provide a complete description of the steps to build the database, its limitations, and the potential products for diverse stakeholders. BENFEP is structured to be analysed with data science tools and geographic information systems software.

## 2 Methods

### 2.1 BENFEP_v1 briefing

The first version of BENFEP (BENFEP_v1) integrates metadata and georeferenced quantitative data of benthic foraminifera species (living, dead, and living plus dead) from surface sediment samples collated from 50 published and unpublished documents released between 1951 and 2022. The number of samples supplied by each publication to the database varies among authors (see Table A1). The database includes samples ranging from 60ºN to 54ºS (Fig. 1) and localized from intertidal waters (0 m water depth) to the deepest curated sample at 7280 m water depth. BENFEP includes 2509 stations, 3077 samples, and 1091 foraminiferal taxa (including species and below species-level designations) plus 400 benthic foraminiferal identifications to genus level.

### 2.2 Data source and selection protocols

The BENFEP_v1 database incorporates entries with georeferenced quantitative data of benthic foraminifera species from the Eastern Pacific surface sediments. We consider data as quantitative when the species abundance in an assemblage is provided as number of individuals (counts), relative abundance (percent) or density (number of individuals per volume unit). A primary source of information for mid to late twentieth-century´ entries was the compilations by Culver and Buzas (1985, 1986, 1987),

Ingle and Keller (1980) and the historical references by Finger (2013). For more recent publications, we used the search engines of Scopus, Journal of Foraminiferal Research, JSTOR and PANGAEA (accessed between early 2020 and March 2022), using the keywords "benthic foraminifera" together with geographic terms such as "Eastern Pacific", or specific geographical terms

of this region, such as "California", "Chile", "Santa Barbara", "Alaska", etc, as well as the authors´ networking. There are 31 documents published between 1929 and 2019 characterizing assemblages of living and dead assemblages of benthic foraminifera from surface sediments in the Eastern Pacific that could not be incorporated in BENFEP_v1 because species assemblage data are provided in graphs, as species presence or range of abundances (e.g. common, rare, abundant). The geolocation of the samples and the authors of those publications can be accessed from

https://doi.org/10.1594/PANGAEA.947114 (Diz et al., 2022a) and they are represented in Figure B1.

A substantial number of entries of BENFEP_v1 come from print-only publications including unpublished theses accessed through universities-interlibrary loans (91%). From these, only 7.6% could be digitized, and the remaining (typewritten or hand-written tables) had to be converted to digital format manually (92.4%). In those cases, entries were doubled or when necessary, tripled checked to minimize errors and as a quality control. Besides, BENFEP_v1 retains the original format in

which census data were published; percentage, counts or densities representing 69, 30.7 and 0.3% of the data respectively. It also includes any non-numeric data used by authors in their original publication to indicate the presence or a non-quantitative value of a particular species (e.g., "x", "<1").

## 2.3. Data geolocation

The samples integrated in BENFEP were georeferenced using the coordinates listed in the original publications. In Smith,

(1964) and Walton (1955), coordinates are not indicated in the original publication along with the benthic census data, and they had to be retrieved from another publication that used the same stations (Smith, 1963; Walton, 1954). For 30.3% of the samples, their location was only shown on maps. In those cases, the maps from the publications were digitized to raster format and georeferenced through ArcGIS software using geographic decimal degrees and World Geodetic System of 1984 (WGS 84 – EPSG:4326). These rasters were then displayed with ArcGIS to extract the sample geolocation by manual digitizing. In those

cases, when the resolution and precision of the map provided in the publication are clearly insufficient, the present coastline was retrieved using high-resolution satellite and aerial world imagery (World Imagery WMS server) and the samples´ geolocation was obtained by combining both sources of data. It is worth mentioning that the coarse resolution of some hand-drawn maps, particularly those published in mid-twentieth- century surveys, might not be totally accurate.

All the obtained geolocations were plotted as point features using a high-resolution satellite and aerial world imagery as a base

map to validate their position. In the cases where the sample location resulted in an inland position, the data was cross-validated and checked, from these analyses there were two possibilities: (i) typing errors in the original source or (ii) land-reclamation activities in the area since the sample was collected. A few samples (11) are not georeferenced because samples´ location is missing from maps (or lists provided by the authors) and other samples (2) are currently located inland.

## 2.4 Taxonomic harmonization

The datasets contributing to BENFEP_v1 come from multiple sources published over the last 70 years, and therefore taxonomic inconsistencies between authors are expected. Aiming to harmonize the spectra of genus and species from the original sources, we standardized the original taxonomy using the currently valid taxonomic assignments of the World Foraminifera Database (Hayward et al., 2022), a part of the World Register of Marine Species (WoRMS). In order to find the valid species name, we searched each author´s original species assignment in the WoRMS research engine. This procedure enables to identify whether

the original species name is accepted (valid species) or if it is a synonymous of the valid species or taxa correspond to a variety or a subspecies. When the original species name was not currently in use, it was substituted by the valid species, subspecies, or variety name. Species names annotated with "cf." or "aff." were not considered as separated species. Some taxa included in

BENFEP_v1 are considered as "fossil only" by WoRMS. Nevertheless, we kept those in the database. There are several reasons to explain the occurrence of a species categorized as "fossil only" in a sample; it represents a true displaced fossil species from ancient sediments (reworking), a mistaken identification, and, an extant species inaccurately attributed as "fossil" by WoRMS. As it is not clear which of these circumstances applies in each case, we decided to maintain the species to prevent information losses in case of future re-evaluation of the "fossil range" by WoRMS. The species identified to genus level with only one species by one author (e.g., Genus A sp.) were assigned to the column name designed by the genera followed by "spp." (e.g., Genus A spp.). However, if an author indicates two or more "sp." species for the same genus (e.g., Genus B sp1, Genus B sp2), a column name with the undetermined species followed by the author´s name is used (e.g., Genus B sp1Golik, thus "Golik" refers to the data set of Golik, 1965). The columns named "Indeterminate calcareous" and "Indeterminate agglutinated" included individuals not identified at the genus or species level in the original publication and included in more general categories such as "other calcareous", "miliolids", "lagenids", or "other agglutinated", respectively. When authors did not provide information about the test nature (e.g., agglutinated, calcareous), census data of the non-identified forms were placed under the column "Indeterminate unknown". The WoRMS search engine was lastly accessed on 22-12-08 using the package "worrms" (Chamberlain, 2020) through R version 4.2.1 (R Core Team, 2022) to obtain updated scientific names, authorities, AphiaID, rank and the species "fossil range" (renamed as "occurrence" in this study). The taxonomic information retrieved from WoRMS, together with the authors´ original assignations and specific remarks on the harmonization procedure are included in the Supplement, File 1, File 2 and File 3, respectively. Extended explanations about some species, in particular, those referring as "potentially fossil" by the original authors, are included in Supplement (File 4). Formal discussions of the taxonomic concepts used by the authors of the publications and by WoRMS are out of the scope of this study.

## 2.5 Structure of the database

The BENFEP_v1 database is provided in short format and long format to reach a high spectrum of final users. The short format (BENFEP_v1_short) consists of 3077 rows and 1565 columns. Each row contains information of one surface sample distributed in metadata (columns 1 to 23 and columns 1556 -1565) aiming to provide all the necessary information for users to assess the quality of the faunal dataset and manage the data at their own convenience. The metadata for each sample were collated from the original source and include information about the publication, name of the research vessel used to collect the sample, sampling year, details regarding different sampling methodologies such as sampling devices and sampling interval (in centimetres at the seabed), format of the quantitative data in which data were originally published (percent, counts, density), type of assemblage (living, dead, and living plus dead), size fraction in which foraminifera were studied, picking and staining protocols to identify living foraminifera, geolocation (latitude and longitude) and water depth of the surface sediment sample. We also included as metadata where we obtained the data from (provided by authors, obtained from machine or manual digitization, or retrieved from repositories), the doi of the dataset when hosted in an open access repository, the source of the geographic coordinates (obtained from tables in the publication or digitized maps). Additionally, in columns 1556-1565 we coded whether the number of counted individuals in each sample is equal to or higher than 100, 200 and 300 individuals. Meaningful annotations regarding the sample entry was spared in seven columns dedicated to the meaning of non-numerical data, comments about some species, assemblage characteristics, volume of the sample (when data are provided in density), size fraction, sample geolocation and others. Benthic foraminifera species quantitative data, one taxon per column is indicated in columns 24 to 1554. The species, varieties and subspecies names are identified in full in one column (e.g., genus and species or genus species and var. or subsp.). A column representing the sum of species abundance per each sample (column "total") was added at the end of the species quantitative data. Users should check the column "format" for indications whether the value in the column represents the sum of percentages, counts or densities. An empty cell in any column indicates that there is no information available. The users of the short format are referred to Supplement (File 1) for comprehensive taxonomic information of each taxa and to Supplement (File 2) for the original authors´ taxonomic concepts.


The long format of BENFEP_v1 (BENFEP_v1_long) contains the 33 columns reflecting the metadata described above for BENFEP_v1_short and three columns describing the harmonized foraminiferal designation ("entity"), each species quantitative data ("abundance") and the total abundance in the sample ("total", see column "format"). This information is followed by the taxonomic information extracted from WoRMS ("valid_authority", "status", "rank", "AphiaID", "kingdom",

"phylum", "class", "order", "family", "genus", "occurrence" ) and each author taxonomic concept ("authors_taxo"). Table C1 and Table C2 detail the meaning of each column and column codes of BENFEP_v1_short and BENFEP_v1_long. The database in its two versions is presented in text format and can be managed with virtually any software.

## 3 Results and discussion

### 3.1 Samples distribution

The sample distribution in BENFEP_v1 is dictated by the availability of, and access to, benthic foraminifera quantitative datasets. The geographic range of samples varies between 60ºN and 54ºS and from 70ºW to 179ºW. The largest density of quantitative data occurs between 40ºN and 30ºN followed by groups of stations centred at 60ºN and between 10ºN and 17ºN (Fig. 1, see also Video Supplement). There are some spatial gaps in benthic foraminifera census data, such as the regions between 17 and 21ºN and several narrow latitudinal intervals in the Southern Hemisphere (40-45, 36-39, 33-35, 29-31ºS). The

water depths range from tidal (0 m) to 7280 m, but 50% of stations are collected between 40 and 550 m of water depth (Fig. 2). From Fig. 1 and Fig. 2, it remains clear that the Eastern Pacific in ocean areas deeper than 3000 m (i.e., lower abyssal zones, following van Morkhoven et al., 1986) are noticeably understudied and that far more studies are needed there to obtain a full overview of benthic foraminiferal distributional patterns. Indeed, the highest number of samples in lower abyssal environments (deeper than 3000 m, Fig. 2) is from the South Pacific and they come from expeditions carried out during the

1960s and 1970s (Bandy and Rodolfo, 1964; Resig, 1981).

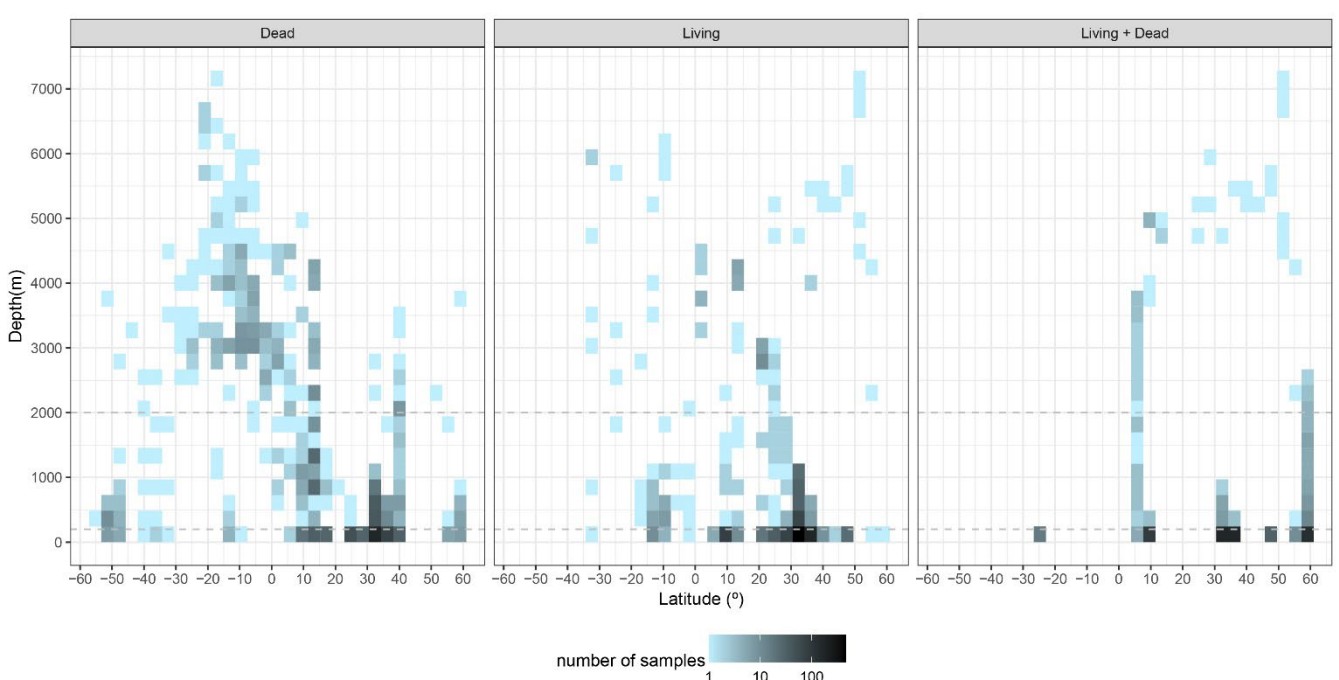

**Figure 2:** Distribution of samples with water depth and latitude. Horizontal dashed lines separate the neritic (0- 200 m), the bathyal (200-2000 m) and, the abyssal zones (>2000 m) following bathymetric divisions of van Morkhoven et al. (1986). The graphs were elaborated with

the package "tidyverse" (Wickham et al., 2019) using R version 4.2.1 (R Core Team, 2022).

**3.2 Research vessels, sampling devices and sampling intervals**

Research expeditions were carried out on board of different Research Vessels; *Velero IV*, *Spencer F. Baird*, *McArthur*, *Yaquina*, *Golden West*, *Atlantics II*, *Puritan*, *Horizon*, *Meteor* are some of the 35 cited research vessels (information taken from "rv_1" and "rv_2", see Appendix C). Alternatively, some samples were provided by miscellanea collections from Scripps
Institution of Oceanography and Allan Hancock Foundation.

Samples were collected using a variety of devices (at least 18 different samplers, Table C1 and Table C2), but most of samples were taken using a gravity corer (20.5%) and Hayward orange peel grabs, Box corer, Phleger corer and miscellanea tools (mostly in shallow water depths), with percentages around 15-8% each (Fig. 3). The most common sediment sampling interval
below the seafloor is 0-1 cm (41.4%), where benthic foraminifera are distributed between dead (9.9%), living (24%) and living plus dead (7.6%) assemblages. Slightly deeper sampling intervals (e.g., 0-2, 0-3, 0-5 cm, etc, Fig. 3) represent 38.4% of the samples in the database (Fig. 3). There are 20.2% of the samples classified as "surface samples", representing authors´ generic assignations to the uppermost centimetre of the sediment (e.g., "surface", "core-top").

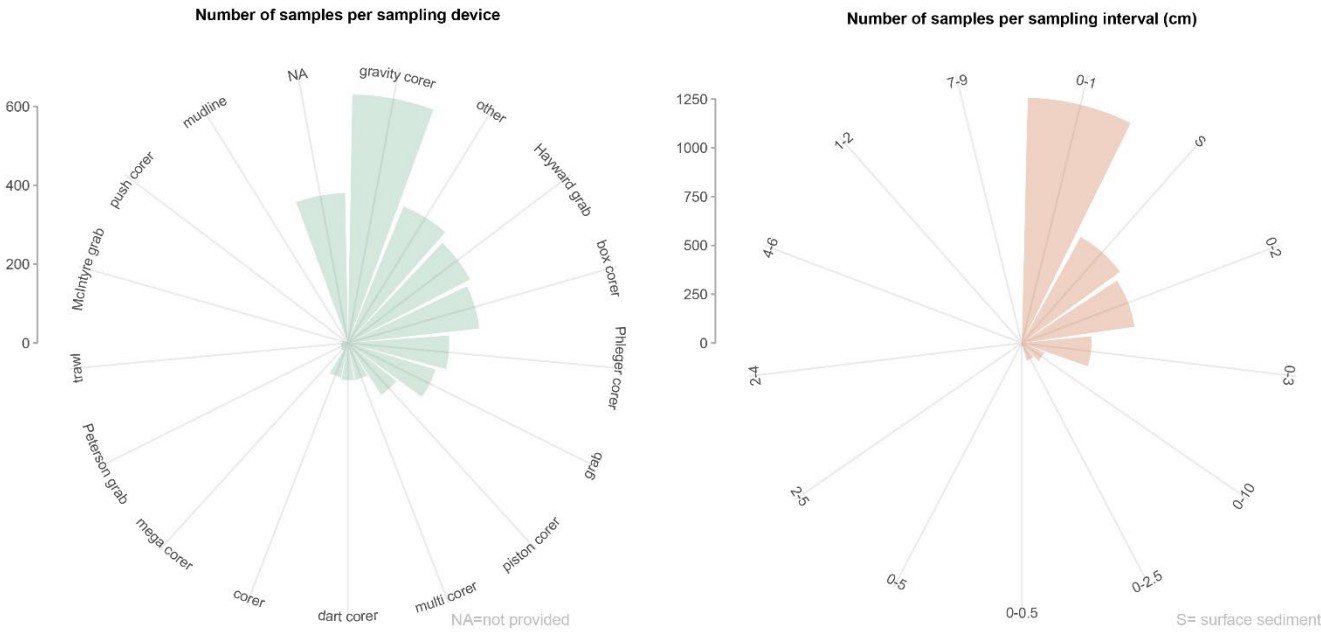


**Figure 3:** Sampling devices and sampling intervals in BENFEP_v1. The distribution of sampling devices is calculated using the "dev_1" column (see Table C1 and Table C2 for more information). The graphs were elaborated with the package "tidyverse" (Wickham et al., 2019) using R version 4.2.1 (R Core Team, 2022).

**3.3 Benthic foraminiferal assemblages**

The BENFEP_v1 database reports data of living (40.1%), dead (33.6%), and living plus dead (26.3%) benthic foraminifera. The Rose Bengal staining (Walton, 1952) is the only method used by authors to distinguish dead (non-stained) from living (stained) foraminifera at the time of sampling. Living plus dead refers to an assemblage were living (stained) and dead (non-stained) are counted together in the same sample. The stain is mixed with different solvents, being the most used formaldehyde (54.8%), followed by alcohol (19.7%) and others, which include seawater and distilled water (25.5%). Samples were mainly
dry picked after flotation (54%) with a density liquid (mostly $Cl_4C$), which was a common practice between 1951 and 1980.

Most of benthic foraminiferal assemblages were analyzed in the smallest size fraction commonly used in benthic foraminiferal studies. For example, 65.4% of the samples were analyzed using 42, 61, 62, 63 and 74 µm as the lower end of

size fraction (e.g. assemblages where studied in the >42 µm, >61µm, >62 µm size fraction, etc.). The 6.1% is using 88 and 105µm and 17.7% the 125, 149, 150, 200, 212 and 500 µm as the lower end of size fraction. The size fraction used for foraminiferal analysis is not reported in 10.8 % of the publications (Table A1 and Fig. 4), which correspond to four entries; Phleger (1965), Landford and Phleger (1973), Bergen and O´Neil (1979) and the historical data reported by McGann (2002).

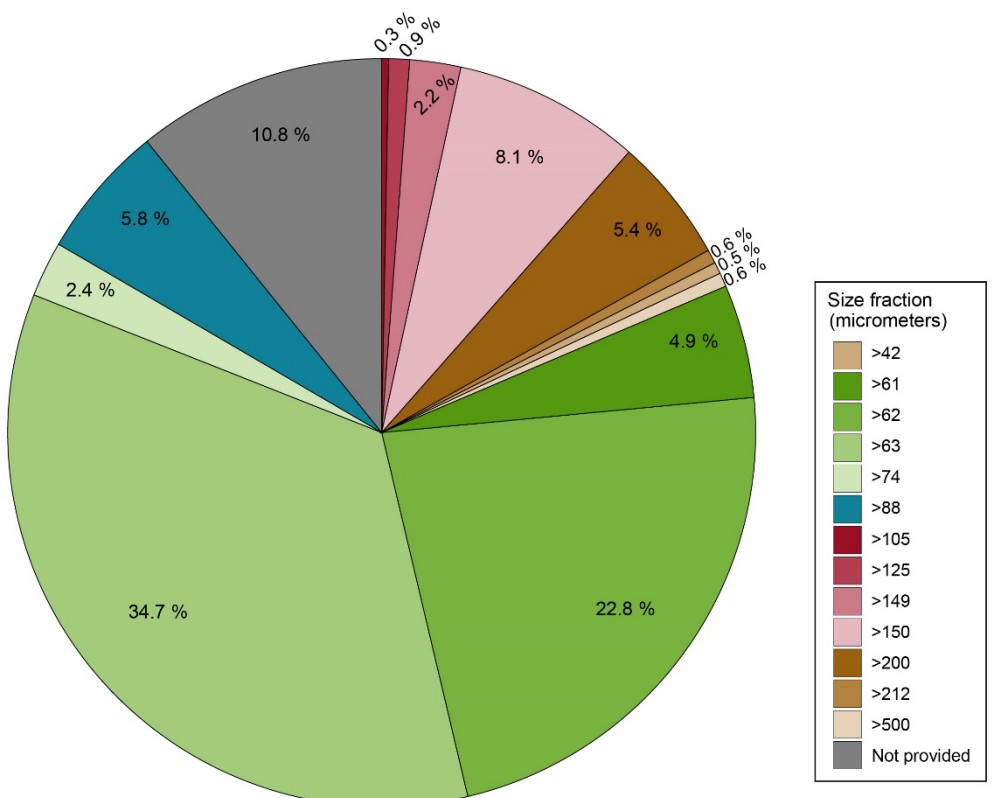

**Figure 4:** Distribution of the size fractions used in the benthic foraminiferal studies included in BENFEP_v1. The graph was elaborated with the packages "ggforce" (Pedersen, 2019) and "tidyverse" (Wickham et al., 2019) using R version 4.2.1 (R Core Team, 2022).

## 3.4 Benthic foraminiferal species

The BENFEP_v1 dataset includes a total of 1091 valid taxa (1073 species, 14 varieties, 4 subspecies) plus two taxa of uncertain status (*Serpula lobata* and *Ammonia avalonensis*) corresponding to 335 foraminiferal genera belonging to the classes; Globothalamea (64%), Tubothalamea (11.3%), Nodosariata (19.6%), Monothalamea (4.8%). In addition to the accepted taxonomic entities, the database contains 400 benthic foraminifera individuals identified to genera level (i.e., "spps"). The genera with the largest number of valid species (excluding subspecies and varieties) is *Bolivina* (46) followed- in decreasing order- by *Quinqueloculina*, *Uvigerina*, *Reophax*, *Fissurina*, *Lagena*, *Bulimina*, (22-32 species, Fig. 5, see also Supplement File 1).

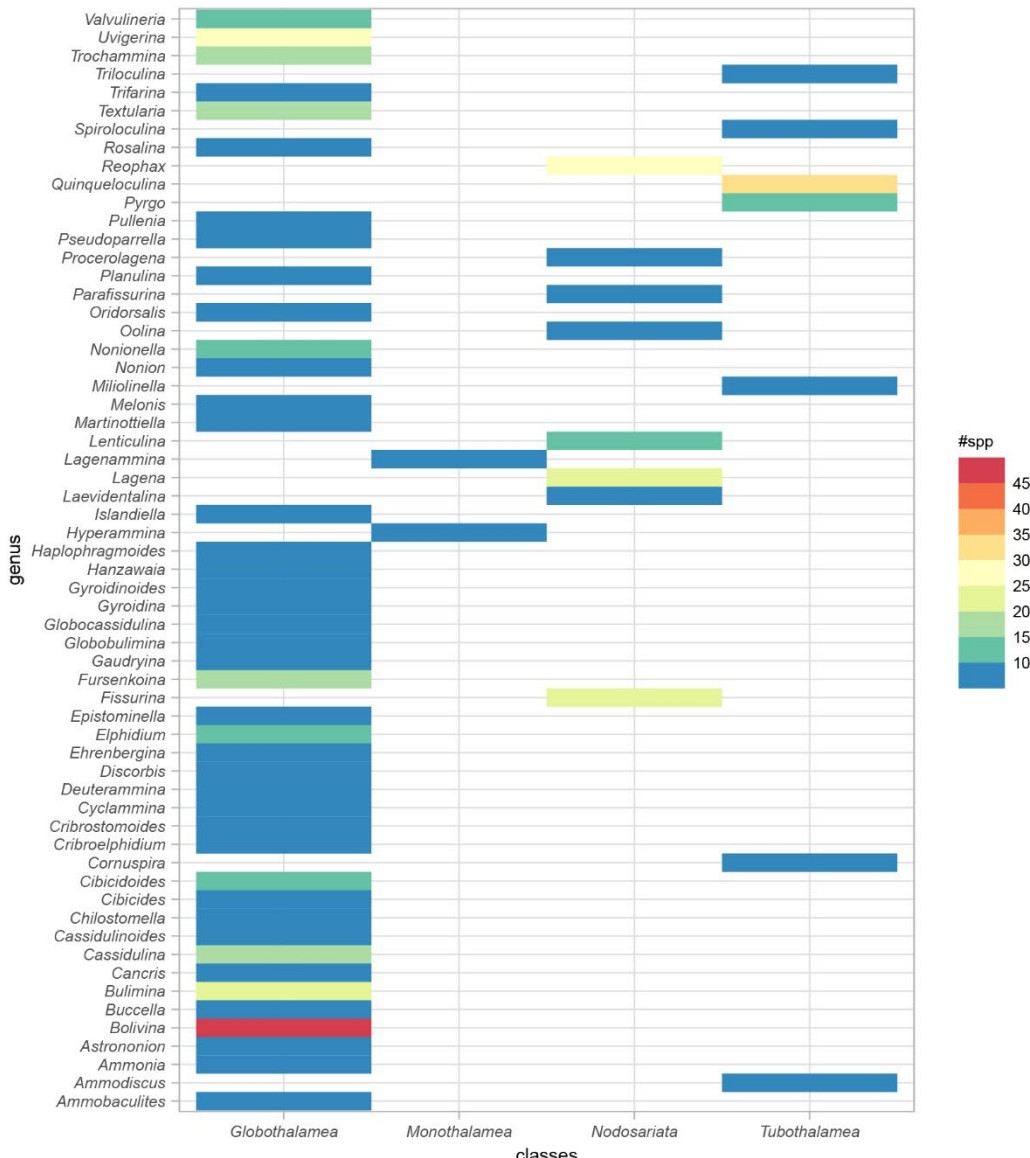

**Figure 5:** Number of valid species per foraminifera genus and its distribution among the classes indicated by WoRMS (Hayward et al., 2022, last accessed on 22-12-08). Only genera with 5 or more species are represented in the figure. The graph was elaborated with the package "tidyverse" (Wickham et al., 2019) using R version 4.2.1 (R Core Team, 2022).

265   The BENFEP_v1 database contains 292 valid species (excluding varieties and subspecies) that can be considered rare, with a mean relative contribution lower than 1% (Murray, 2013; calculation based on samples with counts above 100 individuals analysed in the > 61, >62, >63, >74 and >88 μm size fractions). Furthermore, the highest number of taxa (90) is found in a station studying dead individuals located in the South Pacific at 1800 water depth (Ingle et al., 1980, Fig. 6D). BENFEP_v1 integrates quantitative data across a variety of marine environments, thus, the relative abundance of particular species varies

270   geographically and with water depth (Fig. 6A-B, E). For example, *Textularia mariae*, *Elphidium excavatum* subsp. *clavatum* and *Globocassidulina crassa* are frequent in neritic zone while *Nodulina dentaliniformis* and *Nutallides umbonifer* characterize the abyssal zones (Fig. 6E).

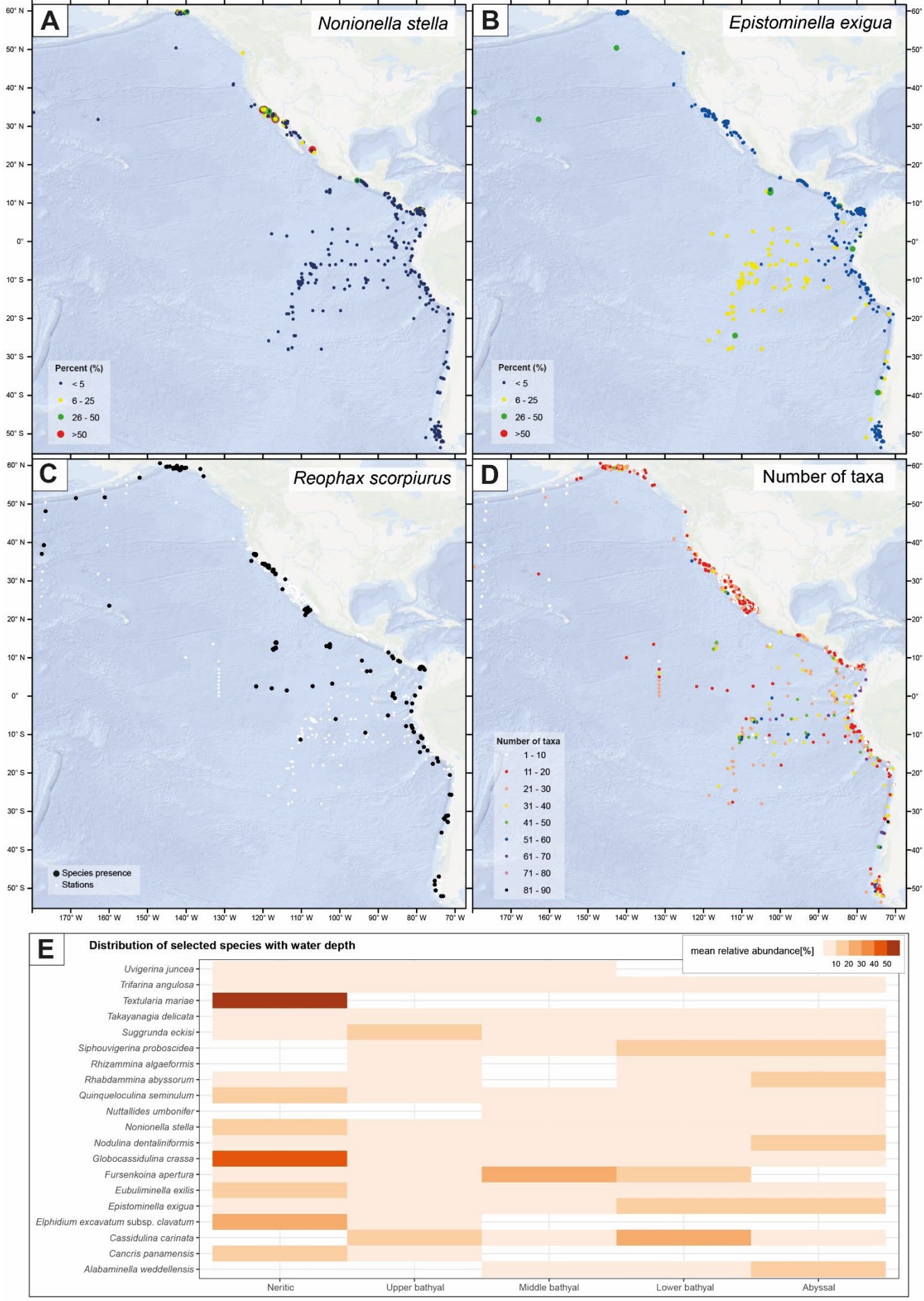

**Figure 6:** Geospatial representation of selected species relative abundance (A, B) and presence (C), total number of taxa (D) and selected species mean relative abundance with water depth (E) in BENFEP_v1. Water depth ranges in figure E are as follows: neritic (0-200 m), upper bathyal (200-600 m), middle bathyal (600-1000 m), lower bathyal (1000-2000 m) and abyssal (>2000 m). The relative abundance of species in figure A, B and E are calculated from a percentage file that integrates samples with counts of more than 100 individuals in the >61, >62, >63, >74 and >88 μm fraction**s**. The calculations of species presence (C) and total number of taxa (D) are calculated integrating the information provided by non-numerical data. The maps of A-D were made using ArcGIS software version 10.8.2. The global relief model integrates land topography and ocean bathymetry (Sources: Esri, Garmin, GEBCO, NOAA NGDC, and other contributions). The graph in E was elaborated with the package "tidyverse" (Wickham et al., 2019) using R version 4.2.1 (R Core Team, 2022).

## 3.5 Potential applications of BENFEP

The high number of stations with benthic foraminifera quantitative data collated from surface sediments of the Eastern Pacific together with the metadata provided, make the BENFEP_v1 database a reference one for a specialized community working on present and past benthic foraminiferal distributions. The database has the potential of being integrated with other databases hosting taxonomic, abundance or biogeographic information of other microfossils, thus serving as a source of ecological information (e.g., biodiversity, ecosystem functioning) for shallow and deep-sea monitoring, management, and conservation (Danovaro et al., 2020). Fig. 6 displays some of the potential applications of BENFEP ranging from the relationship between species and a particular environmental variable (i.e., water depth, Fig. 6E) which can be extended to another, externally accessed environmental variable, the geographic distribution of the relative abundance of species (Fig. 6A, 6B) to species presence (Fig. 6C) and number of taxa (Figure 6D).

## 3.6 Limitations of the database

### 3.6.1 Taxonomic concepts

The species-level taxonomy of benthic foraminifera is mainly based on morphological traits, whose identification criteria might differ among authors, particularly if we consider the time elapsed between some publications. This could represent a limitation which is shared among global or regional databases curating published data from other modern marine microfossil groups (Leblanc et al., 2012; Siccha and Kucera, 2017; Hernández-Almeida et al., 2020). However, the effect of diversified taxonomic concepts might be augmented in benthic foraminifera, whose modern taxa (2400 living species, Murray, 2007) outnumber other marine microorganisms' groups with fossilizing potential, such as planktonic foraminifera (n=50 living species, Brummer and Kucera 2022), coccolithophores (n=200 extant species, Young et al., 2003) or radiolarian, with at least 900 species (Biard, 2022). Despite the effort to harmonize the taxonomy, it is likely that incorporating data from different authors with diverse taxonomic concepts (e.g., there are 499 species identified by a single author) and potential misidentifications (e.g., see Supplement File 4) could have artificially biased the number of species.

### 3.6.2 Data originally sourced in percentage

The data provided in percentage sometimes do not add to 100%. There are several explanations for this. Firstly, the presence of symbols (such as "x", "<0.1") or incomplete assemblage description (e.g., datasets including only species beyond a particular threshold in their relative abundance) necessarily preclude that the sum of the relative contribution of species reaches 100%. We refer users to the "remark_1", "remark_3" section of the database for additional information about the assemblage characteristics (see Table C1 and Table C2). Secondly, rounding of decimals to entire numbers in the original sources might have led to percentages lower or higher than 100%. A few samples from Butcher (1951) contain well above 100%. We hypothesize that they are probably the result of typing errors in the original sources. In any case, we decide to retain quantitative data in their unabridged form because there are potential applications of the database insensitive to percentages such as species presence.

### 3.6.3 Non-numerical data

There are 18 datasets which include non-numerical data ("x", "<1") in their records (see "remark_1"). Those data might
interfere in the calculation of the relative abundances and some diversity indexes (e.g., Shannon Weaver). However, they provide useful information on species presence and therefore they are potentially useful for biogeography and calculations of species richness. General suggestions on how to manage non-numerical data in R can be found in Supplement File 5.

### 3.6.4 The representativeness of the surface sediment assemblages as recent analogues

One of the purported applications of BENFEP_v1 is to provide a quantitative estimate of recent benthic foraminiferal
assemblages that could be later used in palaeoenvironmental interpretations (e.g., Fig. 6). The database integrates quantitative data obtained from oceanic regions with different depositional environments, sedimentation rates, carbonate preservations and types of assemblages, collected over different sampling years and using an array of sampling devices that might result in diversion from recent conditions. For example, dead benthic foraminifera obtained from surface sediments might not be representative of the surface if the sampling device fails to recover the sediment-water interface or sedimentation rates are very
low. The 36% of the surface sediment samples were retrieved using different types of coring devices (gravity, piston, dart and Phleger corer, calculations using "dev_1"), which are sampling techniques that can cause perturbation or miss-sampling of the surface sediment (Weaver and Schultheiss, 1990). Since the studies included in our database did not date the surface sediment (except for Palmer et al., 2020), we cannot discard that some samples correspond to pre-Holocene conditions. The most comprehensive compilation of sedimentation rates from core-top samples is from the equatorial Pacific and shows highly
variable values, ranging from 0.8 to 14.2 cm/ka (Mekik and Anderson, 2018), meaning that surface sediment samples in this region correspond to recent conditions (assuming that no perturbation occurred during sampling). Reworking, downslope transport and carbonate preservation might be other factors influencing the composition of the assemblages obtained from the surface sediments. The presence of "potentially fossil" species reworked from ancient outcrops (see "remark_2" and Supplement File 4) is included in the datasets of Bandy and Arnal (1957), Echols and Armentrout (1980), Ingle et al. (1980)
and Zalesny (1959). Still, they represent less than 5% of the assemblage. The contribution of specimens displaced specimens from shallower locations is also low, as indicated by Bandy and Arnal (1957); Ingle et al. (1980); Harman (1964), Pettit et al. (2013), Uchimura et al. (2017) and Zalesny, 1959. Finally, Pettit et al. (2013) in the Gulf of California, and Boltovskoy and Totah (1987) and Resig (1981) off South America in samples below the carbonate compensation depth, are the only authors mentioning poor preservation of calcareous benthic foraminifera.


BENFEP_v1 includes information of living, dead, and living plus dead assemblages whose suitability for building recent analogues is under discussion among the scientific community. The use of Rose Bengal as "vital" staining could be controversial because attached bacteria or algae, decaying protoplasm of dead individuals might stain, resembling the staining of the protoplasm of a "true" living individual (see review in Schönfeld, 2012). However, it is still the most widely used method
to distinguish "living" (stained) from "dead" (non-stained) foraminifera and it is considered reliable if used cautiously. It might be argued that only living foraminifera should be used to consider baseline studies (Schönfeld, 2012). However, it might also be considered that living assemblages represent a "snapshot" of the foraminifera living at the specific time of sampling and do not hold the time-averaged representativeness of the dead assemblages (Murray, 2000). Regarding all these potential concerns, we have incorporated a rich collection of metadata to BENFEP_v1 that can be used by the final users to evaluate data quality
and to tailor the final output to their specific criteria.

## 4 Recommendations for archiving benthic foraminifera quantitative data

Data sharing in easily accessible formats and public repositories should be the core of the commitment of scientists, universities, and research institutions to open science. Data reusing is not only precluded by lack of data sharing but also by incomplete or lacking metadata, taxonomic information, etc, which are essential to provide the single user or the synthesiser with the information to evaluate the quality of data. In the process of building this database, we have found several issues that we raised as recommendations aiming to encourage best practices in data reporting.

-*Data sharing*. Publishers should commit to FAIR data practices (Wilkinson et al., 2016) and authors must share their published data in a readily accessible format and in public repositories to avoid the irreversible loss of valuable quantitative data. An important disadvantage of machine and manual digitalization is that both are time-consuming and might result in typing errors.-

*Raw data*. Ensuring reproducibility, quality checking and further use of data require raw data, that is, species counts and total counts per each sample. It has been a common practice to provide quantitative data in relative abundance with generic information about the number of individuals counted by sample. As mentioned before, this format is prone to error and hinders, at least, data reusing for some diversity calculations (e.g., rarefaction).

-*Metadata*. Providing detailed information about each station´s sampling device, sampling interval, geographic coordinates, picking, and staining protocols, research vessel, sampling year, etc. It would be avoidable to describe samples´ metadata using unspecific generalizations.

-*Taxonomy*. Providing full taxonomic references of all species. Taxonomic information and supporting images are crucial elements for reliable taxonomic harmonization and data reusability.

## 5 Data availability and future plans

The BENFEP_v1 database can be accessed from https://doi.org/10.1594/PANGAEA.947086 (Diz et al., 2022b). This database is conceived as a springboard to store future quantitative data of benthic foraminifera in the East Pacific and make them available to the scientific community. It will be open for any new quantitative data entry and thus, it welcomes any new data published or provided by any contributor. The database will be updated by the authors once a considerable number of new entries need to be incorporated or changes are required to update taxonomic categories to an existing version. New versions of BENFEP will be submitted and curated in PANGAEA. Collaborations with individual researchers and institutions are welcomed specially regarding potential expansion to other ocean basins.

## 6 Conclusions

We present the BENFEP database, the largest open-access database of quantitative data of benthic foraminifera from surface sediments compiled up to date. BENFEP_v1 contains harmonized census counts of 1091 foraminiferal taxa (including species and below species-level designations) of living, dead, living plus dead benthic foraminifera from 3077 sediment samples, corresponding to 2509 stations of the Eastern Pacific. It also contains a rich collection of metadata gathered from 50 documental sources spanning the last 70 years. BENFEP_v1 prospective is to function as an alive repository for new entries and a reference database for palaeoenvironmental reconstructions, as well as biogeography and biomonitoring studies. The database is friendly coded and can be accessed using different software, aiming to a broad spectrum of users and tailoring needs.

**Appendices**

Appendix A

**Table A1.** Number of samples per contributor, type of assemblage, and size fraction.

| Authors | Living | Dead | Living plus Dead | Fraction (>µm) |
|---|---|---|---|---|
| Bandy and Arnal, 1957 | | 36 | | 61 |
| Bandy and Rodolfo, 1964 | 19 | | | 500 |
| Belanger et al., 2016 | | 27 | | 3 (63), 24 (125) |
| Bergen and O´Neil, 1979 | | | 95 | not indicated |
| Bernhard et al., 1997 | 9 | | | 63 |
| Boltovskoy and Totah, 1987 | | 8 | | 63 |
| Brenner, 1962 | | 81 | | 200 |
| Burmistrova et al., 2007 | | | 16 | 42 |
| Butcher, 1951 | | 78 | | 62 |
| Echols and Armentrout, 1980 | | | 102 | 62 |
| Enge et al., 2012 | 2 | | | 63 |
| Erdem et al., 2020 | 11 | | | 63 |
| Erskian and Lipps, 1977 | | | 44 | 200 |
| Gardner et al., 1984 | | 67 | | 149 |
| Glock et al., 2020 | 8 | | | 63 |
| Goineau and Gooday, 2019 | 11 | 11 | | 150 |
| Golik, 1965 | 85 | | 124 | 63 |
| Harman, 1964 | | 26 | | 61 |
| Heinz et al., 2008 | 7 | | | 63 |
| Hromic et al., 2006 | | 35 | | 63 |
| Ingle et al., 1980 | | 18 | | 61 |
| Lankford and Phleger, 1973 | 102 | | | not indicated |
| Liu, 2001 | | 37 | | 63 |
| Loubere, 1994 | | 66 | | 63 |
| Mackensen and Douglas, 1989 | 3 | | | 125 |
| Mallon, 2011 | 32 | | | 63 |
| McGann, 2002 | 94 | | 175 | 83 (not indicated), 186(150) |
| McGlasson, 1959 | 49 | 71 | | 62 |
| Morin, 1971 | 150 | 166 | | 62 |
| Nienstedt, 1986 | | 45 | | 63 |
| Palmer et al., 2020 | | 5 | | 63 |
| Patarroyo and Martinez, 2021 | | 22 | | 63 |
| Patterson et al., 2000 | 22 | | 31 | 63 |
| Perez-Cruz and Machain-Castillo, 1990 | | 48 | | 63 |
| Pettit et al., 2013 | 6 | 9 | | 63 |
| Phleger, 1964 | 76 | | | 62 |
| Phleger, 1965 | 53 | | | not indicated |
| Resig, 1981 | | 121 | | 63 |
| Scott et al., 1976 | 111 | | 112 | 63 |
| Smith, 1964 | 18 | 18 | | 150 |
| Smith, 1973 | 18 | | 22 | 200 |
| Takata et al., 2016 | | 9 | | 105 |
| Tavera et al., 2022 | | | 17 | 212 |
| Tetard et al., 2021 | | 6 | | 150 |
| Uchimura et al., 2017 | | 24 | | 63 |
| Uchio, 1960 | 151 | | | 77(63), 74 (74) |
| Venturelli et al., 2018 | 9 | | | 63 |
| Walch, 1978 | 10 | | | 62 |
| Walton, 1955 | 179 | | | 88 |
| Zalesny, 1959 | | | 70 | 61 |

Appendix B

**Figure B1.** Spatial distribution of samples in the Eastern Pacific from studies which do not provide quantitative assemblage data. The numbers refer to each author´s dataset. Sample geolocation and metadata can be found in https://doi.org/10.1594/PANGAEA.947114 (Diz
et al., 2022a). The procedure for stations´ georeferencing and column coding follows the indications of sections 2.3 and 2.5. The map was made using ArcGIS software version 10.8.2. The global relief model integrates land topography and ocean bathymetry (Sources: Esri, Garmin, GEBCO, NOAA NGDC, and other contributions).

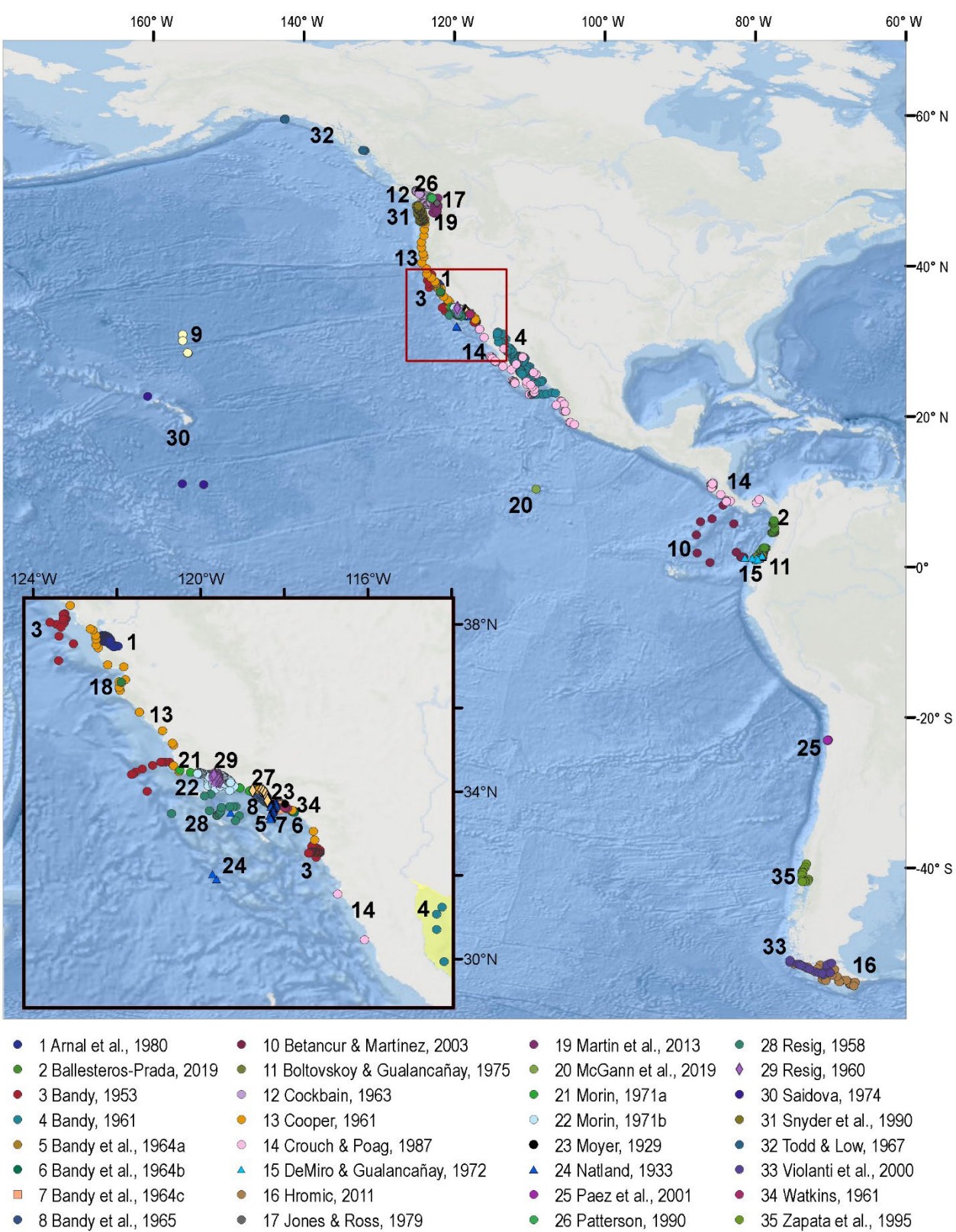

- Arnal et al., 1980
- Ballesteros-Prada, 2019
- Bandy, 1953
- Bandy, 1961
- Bandy et al., 1964a
- Bandy et al., 1964b
- Bandy et al., 1964c
- Bandy et al., 1965
- Berstein et al., 1978

- Betancur & Martínez, 2003
- Boltovskoy & Gualancañay, 1975
- Cockbain, 1963
- Cooper, 1961
- Crouch & Poag, 1987
- DeMiro & Gualancañay, 1972
- Hromic, 2011
- Jones & Ross, 1979
- Martin, 1932

- Martin et al., 2013
- McGann et al., 2019
- Morin, 1971a
- Morin, 1971b
- Moyer, 1929
- Natland, 1933
- Paez et al., 2001
- Patterson, 1990
- Reiter, 1959

- Resig, 1958
- Resig, 1960
- Saidova, 1974
- Snyder et al., 1990
- Todd & Low, 1967
- Violanti et al., 2000
- Watkins, 1961
- Zapata et al., 1995


**Appendix C**

**Table C1.** Explanatory notes on column names and column codes of BENFEP_v1_short.

| Column number | Column Names | Comments | Columns Codes | |
|---|---|---|---|---|
| 1 | authors | Identification code for author or authors of the publication followed by year | See References for a full identification of the publication | |
| 2 | year | Year of the publication | | |
| 3 | source | Source of the data in the database | R | *data obtained from digital repository including an open access repository or a supplementary file in a journal* |
| | | | D | *printed tables in thesis, publications or journal repositories, data were machine digitized* |
| | | | MD | *printed tables in publication or journal repository, data were manually digitized* |
| | | | Author | *data provided by authors* |
| 4 | source_doi | doi of the data source when hosted in an open access repository | | |
| 5 | rv_1 | Research Vessel number 1. This is the main column filled when samples are collected aboard a single research vessel. | Mis | *Miscellanea collections. This applies when the publication does not indicate the Research Vessel but collection of samples from various sources Scripps Institution of Oceanography, Allan Hancock Foundation, oil company* |
| 6 | rv_2 | Research Vessel number 2. This column is filled when samples are collected aboard an additional Research Vessel, different from rv_1 | | |
| 7 | yrv_1 | Sampling year of rv_1. This is the main column filled when data are from rv_1 | | |
| 8 | yrv_2 | Sampling year of rv_2 or different sampling year from yrv_1 | | |
| 9 | yrv_3 | Different sampling year from rv_2 | | |
| 10 | dev_1 | Sampling device used to collect the sediment samples. When several devices are indicated (see dev_2, dev_3), "dev_1" refers to the most frequent | BC | *Box corer* |
| | | | C | *unspecific type of corer* |
| | | | DartC | *Dart corer* |
| | | | FF | *free fall corer,* |
| | | | G | *unspecific type of grab* |
| | | | GC | *Gravity corer* |
| | | | HayG | *Hayward orange peel grab* |
| | | | MC | *Multi corer* |
| | | | McG | *Smith McIntyre grab* |
| | | | MegaC | *Mega corer* |
| | | | Mudline | *Mudline corer* |
| | | | PC | *Piston corer* |
| | | | PG | *Peterson grab* |
| | | | PhC | *Phleger corer* |
| | | | PiC | *Pilot corer* |
| | | | PushC | *Push corer* |
| | | | TC | *Trigger corer* |
| | | | Trawl | *Menzies trawl* |
| | | | Other | *by hand, dredges, skin driving, scoopfish, snapper, tube dragged over the seafloor, Phleger tube* |
| 11 | dev_2 | When filled, it indicates that the authors do not specify the type of the device for each station but they generally indicate the use of two different devices | It applies the same codes as in dev_1 | |
| 12 | dev_3 | When filled, it indicates that the authors do not specify the type of the device for each station but they generally indicate the use of three different devices | It applies the same codes as in dev_1 | |
| 13 | interval | Interval of sediment depth in centimeters | S | *generic designation refering to the surface sediment such as "surface" or "upper few centimetres" or "bottom samples", "modern"* |
| 14 | fraction | Size fraction studied for benthic foraminifera (>micrometers). When necesary, the USA Tyler mesh screen is converted to micrometers | | |
| 15 | assemblage | Type of benthic foraminiferal assemblage | L | *Living (Rose Bengal stained) assemblage.* |
| | | | D | *Dead (un-stained) assemblage* |
| | | | LD | *Living plus Dead assemblage. The abundance of living plus dead foraminifera are combined in the same sample.* |
| 16 | rosebengal | All living assemblages in the database are studied using the Rose Bengal staining method mixed with different solvents | Alcohol | *ethanol, ethyl alcohol, methanol, isopropyl alcohol, unspecific alcohol* |
| | | | Formaldehyde | *buffered formaldehyde* |
| | | | Other | *seawater, glutaraldehyde, distilled water* |
| 17 | picking | Method of picking the foraminifera | Dry | *dry picking after sieving* |
| | | | Wet | *wet picking after sieving* |
| | | | Flotation | *dry picking after using $Cl_{4C}$ flotation method* |
| 18 | format | Format in which the original assemblage data are provided | Percent | *Part in a hundred* |
| | | | Counts | *Number of individuals* |
| | | | Density | *Counts per volume unit* |
| 19 | s_coord | Source of the geographic coordinates | Listed | *listed in the publication* |
| | | | Map | *extracted from the digitized maps provided in the publication* |
| 20 | station | Station identification. For stations described only by a number, we added the surname of the first author of the publication ahead of the station name followed by underscore | | |
| 21 | long | Longitude in degrees from 0 to 180(-180) with positive (negative) values indicating east (west) | | |
| 22 | lat | Latitude in degrees from 0 to 90(-90), positive (negative) indicates latitude north (south) | | |
| 23 | depth | Water depth in meters. When necessary, fathoms or feet are converted to meters by multiplying by 1.8288 or dividing by 0.3048, respectively | | |
| 24-1554 | | valid taxa following WoRMS (last accessed on 22-12-08) or genus asignation. When an author identifies one or more "sps" per genus, the name of the author is indidated after "sp" | | |
| 1555 | total | Sum of colums from 24-1554. See also format column | Columns from 24 to 1554 are valid taxa following WoRMS (last accessed on 22-12-08) or genus | |
| 1556 | n100 | It indicates whether sample counts are equal to or higher than 100 individuals | Yes | *sample counts are equal to or higher than 100 individuals* |
| | | | No | *sample counts are lower than 100 individuals* |
| | | | NC | *the counts per sample are not indicated, however the authors indicate in the publication that samples contain more than 100 individuals* |
| 1557 | n200 | It indicates whether sample counts are equal to or higher than 200 individuals | Yes | *sample counts are equal to or higher than 200 individuals* |
| | | | No | *sample counts are lower than 200 individuals* |
| | | | NC | *the counts per sample are not indicated, however the authors indicate in the publication that samples contain more than 200 individuals* |
| 1558 | n300 | It indicates whether sample counts are equal to or higher than 300 individuals | Yes | *sample counts are equal to or higher than 300 individuals* |
| | | | No | *sample counts are lower than 300 individuals* |
| | | | NC | *the counts per sample are not indicated, however the authors indicate in the publication that samples contain more than 300 individuals* |
| 1559 | remark_1 | Relevant additional information regarding the authors' dataset. This column is dedicated to explanations about non-numerical data | | |
| 1560 | remark_2 | Relevant additional information regarding the authors' dataset. This column is dedicated to comments about species | | |
| 1561 | remark_3 | Relevant additional information regarding the authors' dataset. This column is dedicated to explanations about assemblage characteristics | | |
| 1562 | remark_4 | Relevant additional information regarding the authors' dataset. This column is dedicated to explain the unit of volume in case of the format of the data is density | | |
| 1563 | remark_5 | Relevant additional information regarding the authors' dataset. This column is dedicated to explain size fraction conversions or size related issues | | |
| 1564 | remark_6 | Relevant additional information regarding the authors' dataset. This column is dedicated to explain geolocation-related issues | | |
| 1565 | remark_7 | Relevant additional information regarding the authors' dataset. This column is dedicated to mention issues which do nof fall into the categories of remark_1-6 | | |
| | | An empty cell in any column indicates that there is no information available | | |


**Table C2.** Explanatory notes on column names and column codes of BENFEP_v1_ long.

| Column number | Column Names | Comments | | Columns Codes |
|---|---|---|---|---|
| 1 | authors | Identification code for author or authors of the publication followed by year | | See References for a full identification of the publication |
| 2 | entity | Valid taxa name following WoRMS (last accessed on 22-12-08) or genus asignation | | When an author identifies two or more "sps" per genus, the name of the author is indiated after "sp" |
| 3 | abundance | Quantitative data of the species (entity) abundance. The format of original data is provided in the format column. | | |
| 4 | year | Year of the publication | | |
| 5 | source | Source of the data in the database | R | *data obtained from digital repository including an open access repository or a supplementary file in a journal* |
| | | | D | *printed tables in thesis, publications or journal repositories, data were machine digitized* |
| | | | MD | *printed tables in publication or journal repository, data were manually digitized* |
| | | | Author | *data provided by authors* |
| 6 | source_doi | doi of the data source when hosted in an open access repository | | |
| 7 | rv_1 | Research Vessel number 1. This is the main column filled when samples are collected aboard a single research vessel. | Mis | *Miscellanea collections. This applies when the publication does not indicate the Research Vessel but collection of samples from various sources Scripps Institution of Oceanography, Allan Hancock Foundation, oil company* |
| 8 | rv_2 | Research Vessel number 2. This column is filled when samples are collected aboard an additional Research Vessel, different from rv_1 | | |
| 9 | yrv_1 | Sampling year of rv_1. This is the main column filled when data are from rv_1 | | |
| 10 | yrv_2 | Sampling year of rv_2 or different sampling year from yrv_1 | | |
| 11 | yrv_3 | Different sampling year from rv_2 | | |
| 12 | dev_1 | Sampling device used to collect the sediment samples. When several devices are indicated (see dev_2, dev_3), "dev_1" refers to the most frequent | BC | *Box corer* |
| | | | C | *unspecific type of corer* |
| | | | DartC | *Dart corer,* |
| | | | FF | *free fall corer,* |
| | | | G | *unspecific type of grab* |
| | | | GC | *Gravity corer* |
| | | | HayG | *Hayward orange peel grab* |
| | | | MC | *Multi corer* |
| | | | McG | *Smith McIntyre grab* |
| | | | MegaC | *Mega corer* |
| | | | Mudline | *Mudline corer* |
| | | | PC | *Piston corer* |
| | | | PG | *Peterson grab* |
| | | | PhC | *Phleger corer* |
| | | | PiC | *Pilot corer* |
| | | | PushC | *Push corer* |
| | | | TC | *Trigger corer* |
| | | | Trawl | *Menzies trawl* |
| | | | Other | *by hand, dredges, skin driving, scoopfish, snapper, tube dragged over the seafloor, Phleger tube* |
| 13 | dev_2 | When filled, it indicates that the authors do not specify the type of the device for each station but they generally indicate the use of two different devices | | It applies the same codes as in dev_1 |
| 14 | dev_3 | When filled, it indicates that the authors do not specify the type of the device for each station but they generally indicate the use of three different devices | | It applies the same codes as in dev_1 |
| 15 | interval | Interval of sediment depth in centimeters | S | *generic designation refering to the surface sediment such as "surface" or "upper few centimetres" or "bottom samples", "modern"* |
| 16 | fraction | Size fraction studied for benthic foraminifera (>micrometers). When necesary, the USA Tyler mesh screen is converted to micrometers | | |
| 17 | assemblage | Type of benthic foraminiferal assemblage | L | *Living (Rose Bengal stained) assemblage.* |
| | | | D | *Dead (un-stained) assemblage* |
| | | | LD | *Living plus Dead assemblage. The abundance of living plus dead foraminifera are combined in the same sample.* |
| 18 | rosebengal | All living assemblages in the database are studied using the Rose Bengal staining method mixed with different solvents | Alcohol | *ethanol, ethyl alcohol, methanol, isopropyl alcohol, unspecific alcohol* |
| | | | Formaldehyde | *buffered formaldehyde* |
| | | | Other | *seawater, glutaraldehyde, distilled water* |
| 19 | picking | Method of picking the foraminifera | Dry | *dry picking after sieving* |
| | | | Wet | *wet picking after sieving* |
| | | | Flotation | *dry picking after using $Cl_{4C}$ flotation method* |
| 20 | format | Format in which the original assemblage data are provided | Percent | *Part in a hundred* |
| | | | Counts | *Number of individuals* |
| | | | Density | *Counts per volume unit* |
| 21 | s_coord | Source of the geographic coordinates | Listed | *listed in the publication* |
| | | | Map | *extracted from the digitized maps provided in the publication* |
| 22 | station | Station identification. For stations described only by a number, we added the surname of the first author of the publication ahead of the station name followed by underscore | | |
| 23 | long | Longitude in degrees from 0 to 180(-180) with positive (negative) values indicating east (west) | | |
| 24 | lat | Latitude in degrees from 0 to 90(-90), positive (negative) indicates latitude north (south) | | |
| 25 | depth | Water depth in meters. When necessary, fathoms or feet are converted to meters by multiplying by 1.8288 or dividing by 0.3048, respectively | | |
| 26 | total | Sum of abundance per each station. The format of the original data is provided in the format column. | | |
| 27 | n100 | It indicates whether sample counts are equal to or higher than 100 individuals | Yes | *sample counts are equal to or higher than 100 individuals* |
| | | | No | *sample counts are lower than 100 individuals* |
| | | | NC | *the counts per sample are not indicated, however the authors indicate in the publication that samples contain more than 100 individuals* |
| 28 | n200 | It indicates whether sample counts are equal to or higher than 200 individuals | Yes | *sample counts are equal to or higher than 200 individuals* |
| | | | No | *sample counts are lower than 200 individuals* |
| | | | NC | *the counts per sample are not indicated, however the authors indicate in the publication that samples contain more than 200 individuals* |
| 29 | n300 | It indicates whether sample counts are equal to or higher than 300 individuals | Yes | *sample counts are equal to or higher than 300 individuals* |
| | | | No | *sample counts are lower than 300 individuals* |
| | | | NC | *the counts per sample are not indicated, however the authors indicate in the publication that samples contain more than 300 individuals* |
| 30 | remark_1 | Relevant additional information regarding the authors' dataset. This column is dedicated to explanations about non-numerical data | | |
| 31 | remark_2 | Relevant additional information regarding the authors' dataset. This column is dedicated to comments about species | | |
| 32 | remark_3 | Relevant additional information regarding the authors' dataset. This column is dedicated to explanations about assemblage characteristics | | |
| 33 | remark_4 | Relevant additional information regarding the authors' dataset. This column is dedicated to explain the unit of volume in case of the format of the data is density | | |
| 34 | remark_5 | Relevant additional information regarding the authors' dataset. This column is dedicated to explain size fraction conversions or size related issues | | |
| 35 | remark_6 | Relevant additional information regarding the authors' dataset. This column is dedicated to explain geolocation-related issues | | |
| 36 | remark_7 | Relevant additional information regarding the authors' dataset. This column is dedicated to mention issues which do nof fall into the categories of remark_1-6 | | |
| 37 | valid_authority | Authors of the original described species | | |
| 38 | status | The status of the taxa as indicated in WoRMS. Last accessed on 22-12-08 | | accepted; alternate representation, taxon inquirendum |
| 39 | rank | Taxonomic rank | | Genus; Phylum; Species; Subspecies; Variety |
| 40 | AlphaID | Aphia ID number | | |
| 41 | kingdom | | | |
| 42 | phylum | | | |
| 43 | class | | | |
| 44 | order | | | |
| 45 | family | | | |
| 46 | genus | | | |
| 47 | ocurrence | The occurrence of the taxa in the geological record | | designation of the "fossil range" as stated in WoRMS: fossil_only; recent_only; recent_and_fossil |
| 48 | authors_taxa | The original authors' taxonomic concept for each species | | |
| | | An empty cell in any column indicates that there is no information available | | |

**Video supplement.** Accumulative timeline heatmap showing the geographic distribution of samples´ density in BENFEP_v1. The type of assemblage (dead, living, and living plus dead) is identified using black, red, and green filled circles, respectively. The global relief model integrates land topography and ocean bathymetry (Sources: Esri, Garmin, GEBCO, NOAA NGDC, and other contributions). The slides for the video were made using QGIS software and the video assembly was done with Adobe Premiere software. The video supplement can be accessed from https://doi.org/10.5281/zenodo.7472278.

**Supplement.** The supplement contains 5 files. File 1 indicates the systematics of benthic foraminiferal species listed in BENFEP_v1 following the concepts of the World Foraminifera Database (Hayward et al., 2022, last accessed on 22-12-08). File 2 lists the original authors´ species designations for the species harmonized in BENFEP_v1 and indicated in File 1. File 3 contains specific remarks on the harmonization procedure. File 4 indicates extended explanations about some species. File 5 provides general suggestions on how to manage BENFEP_v1_short in R.

**Author contributions.** IHA conceptualized the study. PD was responsible for metadata collating, benthic foraminifera curation and species harmonization. PD and VGG cross-validated entries of manually digitized data carried out by VGG. RGV and AO were responsible for georeferencing stations and geographic data visualizations while PD was involved in organizing quantitative data for figures and statistics. All authors contributed to designing the structure of the database and actively participated in organizing the manuscript outline, writing, and editing the manuscript at its various stages.

**Competing interests.** The authors declare that they have no conflict of interest.

**Acknowledgements.** The authors would like to acknowledge the efforts of the personnel of the Interlibrary Loan services of the Universidade de Vigo and ETH Zürich to provide us with access to legacy documents. We also extend our acknowledgments to the librarians of Florida State University, University of California Los Angeles, University of California San Diego, University of Southern California, Stanford University and Martin Luther King University for providing access to parts of unpublished PhD thesis or Master thesis.

**Financial support.** VGG was partially funded by Axencia Galega de Innovación, Xunta de Galicia (GAIN) through the program GRC-ED431C 2017/55 to XM1 research group.

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
