# Peer review of "BENFEP, a quantitative database of BENthic Foraminifera from surface sediments of the Eastern Pacific"

_Earth System Science Data, 2022_

## Referee Comment (RC3)

Overall, I greatly appreciate this manuscript, including the inclusion of extensive metadata, representing a large amount of work; I think that this manuscript and its product (BENFEP) provide a great service to the community and should be published. In general, the figures read well and are illustrative. I have some remarks, including some suggesting improvements, some of which are rather minor and nitpicky, but will improve easy of reading and database use by the community.

Remark 1:  In my opinion research on paleoceanographic use of benthic foraminifera is indeed (as stated by the authors of this manuscript) greatly hampered by the lack of integration of (the many) published data, which at least in part is caused by the problems on integrating taxonomy. In this manuscript the authors 'unify' the taxonomy to the WORMS database, which is a good first goal. However, the authors do not evaluate the taxonomic concepts of different authors, and do not aim to do so. In fact, that task might be difficult to impossible, because none of the compiled papers provides good descriptions and/or figures of all taxa mentioned. This database does not include descriptions and/or figures of the taxa, thus the user remains uncertain about the uniformity of the species concepts included in this study. I would think that definitely some of the more 'difficult taxa', e.g., various species names of *Gyroidina/ Gyroidinoides*, of *Lagena*, of *Lenticulina* could well be used differently by different authors, a problem not (fully) resolved by the synonymies given in WORMS. One also cannot exclude authors making mistaken identifications. As an example, *Nuttallinella florealis* White has been extinct since the end Paleocene benthic foraminiferal extinction, thus is unlikely to have been found in surface samples, and was probably misidentified by Liu, 2001. *Obesopleurostomella brevis* and many other uniserial taxa with a complex aperture became extinct in the mid-Pleistocene benthic foraminiferal extinction event (Hayward et al., 2012, Cushman Foundation Foram Res vol 43), and have been marked as 'fossil only' in WORMS. Taxa with such a relatively recent extinction date have been found in surface samples by e.g., Brady in the Challenger Report: see Hayward and Kawahata, 2005, Extinct foraminifera in Brady's *Challenger Report*, J Micropalont. 24, 171-175, who list *O. brevis* as such, as well as *P. alternans*, also listed in the Supplement. This of course might be caused by the age of the surface-sediment sample not being not well controlled (see below, Remark 2). The authors should in my opinion say something about the possibility of mistaken identification, possibly where they are discussing taxa mentioned by one author only (lines 244-on), or where they are discussing 'taxonomic concepts (section 3.6.1). In my opinion, this lack of detailed checking of the taxonomy definitely does not invalidate the present manuscript. The goal of comparing data from different authors (thus taxonomic 'schools of thought'), as well as different collection methods is a very good and worthwhile first step: after all, the Paleobiology Database is extremely useful for the community without all taxonomic confusion having been solved. In fact, this database effort will make taxonomic evaluation in the future easier to do since so many disparate datasets are here collated. However, I think that the authors should mention the potential problems due to taxonomic misidentification and non-realized synonymies.

Remark 2: The apparent occurrence of various extinct species in the database: I suggest that the authors mark it in the Supplement (Species list) if the species is given as 'fossil only' in WORMS, in order to warn the reader/user. In such a case, the explanation could be one or

more of the following: the species may be misidentified (would be interesting to know whether it was tagged as 'living'), the species may not be extinct (in disagreement with WORMS), or the surface sample may in fact not be recent (or even Holocene) in age. Many of the sampling methods described will not recover 'undisturbed' sediment, i.e., the sediment/water interface will not be recovered in most samplers except multicorers. Could the authors say something about this, either in the text or in the description of sample methods? They mention this in line 296, but could they have added a bit more information, e.g., as described above?
It would e.g., be good to know if the authors of the cited article did radiocarbon dating to determine a Holocene age. It would be specifically of interest to know if species described as 'fossil only' in WORMS had been stained and recognized as 'living'.

Remark 3: I am not sure whether I am convinced by the authors that the eastern equatorial Pacific should get priority for such a compilation (maybe there are more data for the Atlantic?). However, the question whether this is the highest priority region on Earth for such a compilation - which could be made because we are looking at a large part of the largest ocean on Earth - is unimportant. After all, we must start somewhere with a global database, and exactly where we start does not really matter - as long as the database is open for extension in the future (see below).

Remark 4: The authors mention that BENFEP is indeed open to such expansion (e.g., lines 23-24), and my earlier remarks indicate that it would indeed be important for the community if BENFEP can be expanded. However, the potential pathway(s) to such extension, e.g., into different oceans, adding additional sources (e.g., articles by Saidova published in Russian), is not very clearly spelled out. How do the authors envisage to keep BENFEP going and growing? Are they proposing to supervise and/or impose quality control on the database? will there be an opportunity for community input? Which entity (university? museum? association?) will; hoist the database in the long term? Potential cooperation which groups hosting databases for other microfossils?

Notes by line number:
23-24: the authors say ' *We complement BENFEP with an additional database integrating metadata and stations geolocation of benthic foraminiferal studies dearth of quantitative data*'. I do not understand the English in this sentence. I see in lines 93-94 and section 4 (lines 305-on) that this is a database containing qualitative record, but this sentence does not explain that. Maybe the authors want to say that they add these because there are so few quantitative datasets? What data does this database contain: presence -absence, or such data as 'rare, few, common, abundant'?

36-37: for the eastern equatorial Pacific, I think that the statement ..'*ongoing deoxygenation..induced by coastal eutrophication*' is incorrect or at least incomplete. After all, the eastern equatorial Pacific is the location of the largest open-ocean Oxygen Minimum Zone not linked to coastal eutrophication (see e.g. Breitburg et al., 2018, Declining oxygen in the global ocean and coastal waters. Science 359, eeam7240). Comparison of figure 1 and the

figure in Breitburg et al shows clear overlap of the studied sites with that open ocean OMZ, and the authors refer to deep-water deoxygenation in line 53.

164-165: The authors say: *'see Supplement for full species description*'. Either I am missing something, or the supplement contains a full list of species, but there is no description of species (i.e., description of their morphology). Please change this text to explain what actually is in the Supplement.

181-190: Text and Caption figure 2. why use words such as 'epi**pelagic**', mesopelagic, bathypelagic, etc.? I think that the words 'xxx-pelagic' are defined for planktonic and nektonic organisms, not for benthos. For example, in general (and in oceanographic textbooks) benthic organisms are described as bathyal or hadal, planktic organisms as bathypelagic or hadal pelagic. The cited paper by Costello & Breyer does not use these xxxpelagic terms for benthos, they use them for planktonic organisms, and for benthics they give the not very useful word 'deep-sea' for benthos below 2000 m. Benthic foram users commonly follow van Morkhoven et al., 1986 depth zones:

**< 200 m depth**: **Neritic**
- coastal: 0–30 m,
- inner neritic: 30–50 m,
- middle neritic: 50–100 m
- outer neritic; 100–200 m

**200-2000 m depth: bathyal**
- 200-600: upper bathyal
- 600-1000: middle bathyal
- 1000-2000: lower bathyal

**>2000 m depth: abyssal**
- 2000-3000: upper abyssal
- >3000 m: lower abyssal

194: delete 'being'.

213-217: maybe add a remark that nowadays Rose Bengal staining is not thought to be very reliable as indicator of actually living specimens at the time of collection?? e.g. Bernhard, J., 2006, Comparison of two methods to identify live benthic foraminifera: A test between Rose Bengal and CellTracker Green with implications for stable isotope paleoreconstructions, Paleoceanography 21, PA4210 states: *'On average, less than half the Rose Bengal–stained foraminifera were actually living when collected'*. In my opinion the authors thus should add some information on the concept of 'stained' vs. 'living.

222-225: The definition of studied grain sizes is confusing. The text states: *For example, 62.5% of the samples were analysed in the >61-74 µm size fraction, 6% in the > 88-106 µm size fraction and 10.5% in the > 125-150 µm size fraction*. I think this must mean that the lower boundary of the studied grainsize is in the stated interval, e.g. ' *the >61-74 µm size fraction'* means that the

full range of grainsizes >63 μm was investigated (as listed in Table A1). I think this must be so, because that is what most people do (e.g. Loubere 1994). However, the text appears to read either as if only specimens in between 61 and 74 μm were studied, or as if specimens >71 μm were studied, either of which is incorrect. Similarly, people study either the 125 μm or >150 μm size fraction, but definitely do not limit themselves to specimens between 125 and 150 μm. Then the caption of Figure 5 shows different numbers again, i.e. instead of >63 μm or >61-74 μm the piechart shows (60, 80) (which I think is the same size fraction). Please standardize this to make sure that the reader understands what is listed unequivocally.

234: 'valid taxa' - I assume this includes subspecies, varieties and such? could the authors add the number of species? below-species level groups may be included in the species by other researchers, thus making species diversity lower. Again, note that some of the species in this database are listed as extinct, and might have been misidentified. The authors mention how many of the species in their list are 'rare' according to Murray 2013. Could they say a bit more about the total number of species - how many of the more common ones are overlapping with Murray?

235: please provide reference for these Classes- who defined these?

248: are these all (90) at the species level or including below species level taxa?

251: typo in Nuttallides (spelled with two letters t)

Figure 7: figurer 7D says 'Biodiversity' but plots number of species. the Number of species is NOT how diversity is defined, it is 'species richness'.

256: what is a 'heatmap'?

261: 'made'  or 'drawn' rather than 'elaborated'.

286: rounding could lead to both >100 as well as < 100%?

316-on (section 5): here the authors should have mentioned that presently many journals require making data available, and will not publish papers without such data, which must be provided not just in the journal, but in an accepted database such as Pangaea. They are escribing some of the broadly accepted FAIR data practices: Findable, Accessible, Interoperable, and Reusable. Their text here should have pointed this out: in many cases 'authors should not be encouraged' - they will be required to do so by reputable journals.  - https://www.nature.com/articles/sdata201618

Table A1: for studies describing 'living + dead', it is possibly to provide the information how many of these species were observed as 'living', how many as 'dead' (with overlap, some species could be present both living and dead)? it would be good information to know whether species were present both as living and as dead.

Table B1:

I think there are mistakes in N200, N300,which are both described as ' It indicates whether sample counts are equal to or higher than 100 individuals'. I think this should be 200 and 300, respectively.

Note to supplement:

There are two entries for *Oridorsalis tener* (681, 682). meant or mistake?

---

## Referee Comment (RC4)

This is a review for "BENFEP, a quantitative database of BENthic Foraminifera from surface sediments of the Eastern Pacific" by Diz et al. BENFEP is, as indicated by the title of the paper, a quantitative database of the distribution of recent benthic foraminifera in the Eastern Pacific. The authors compiled a huge amount of datasets, published since the 1950s, into a comprehensive database. The data has been corrected for species synonyms according to the latest taxonomy published in WORMS. The database includes 1072 fully described species plus 391 benthic foraminiferal entities classified at genus level from 2572 sampling locations. This is a huge database and I have the deepest respect for the amount of work that went into such a piece of work. I can imagine a huge amount of scientific possibilities that will be opened by having access to such a database. First of all, it could be used for calibrating proxies, based on benthic foraminifera assemblages. One example, which directly comes into my mind, is a study like Tetard et al., 2021 regarding foraminifera assemblages on the way to a quantitative oxygen proxy. In addition, this dataset can be used to estimate foraminiferal biomass distribution in the sediments, it can be used to upscale individual metabolic rates to total biogeochemical fluxes and much more. For example, it might be possible to estimate foraminiferal respiration rates in sediments.

Unfortunately, I cannot really access the database files on Pangaea. Even, if I log in, I get the message that access is forbidden due to the data moratorium (Error 403). I shortly screened the other reviews and it seemed that they did not experience this issue. So my review is purely based on the background information in the paper and I cannot review in any way the content of the database. I rated the data quality "good" because I had to give a grade in the review system.

In my opinion, based on the background information in the paper, this database will be of very high interest for a large scientific community including micropaleontologists that try to solve taxonomic issues, paleoceanographers that calibrate new proxies and biogeochemical modelers, who try to include biogeochemical flux rates from activities of different organisms into their models. I have some suggestions that may improve the accessibility of the data and some questions about the future of this database and would only suggest minor revisions for the accompanying paper. As mentioned above, at the moment it seems impossible for me to access the database via PANGAEA, so I cannot review the content of the database itself.

Points of revision for the paper:

Line 121-122: "The whole database can be managed using R version 4.2.1 (R Core Team, 2022). It can be uploaded and managed with geographic information system software such as QGIS and ArcGIS after changing the table format from wide to long."

It would be great, if the database could be managed with geographic information system software such as ArcGIS. If I would have access to the database, I would have tested this myself. I think it would be really helpful, if the authors provide some more information on how to do this. For example, either the database could be uploaded in different file formats. At the moment the file that is uploaded at Pangaea seems to be an .xlsx file. This is of course great for standard users of .xls based software but makes it more complicated for other users (R for example), especially with such a big file. One solution could be to upload the database in different file versions. One text file in long format that could be directly imported into QGis or ArcGIS, another one in wide format for other applications and the original .xslx file. This would be very helpful for users but also would make updating the database in the future very tedious, since all the files would have to be exchanged.

Another possibility would be to publish an R friendly text file that could easily be imported into .xls based software. If this file is in a wide format, it would be easy to provide an R script to convert it into a QGIS/ArcGIS friendly long format, either in the paper or in the supplement. It is also not directly clear

to me, which columns would have to be merged to make the format GIS friendly. Here is an example for such a cookbook (using the gather() argument from the tidyr library):

*benfep_wide*

```
**>   station lattitude longitude uvigerina_peregrina hoeglundina_elegans cibicides_wuellerstorfi**
**> 1       1    10°56'    178°23'                17.3             1000.7                     0.1**
**> 2       2    12°02'    175°59'              5273.2               50.0                      42**
```

```
library(tidyr)
**The arguments to gather():**
**- data: Data object**
**- key: Name of new key column (made from names of data columns)**
**- value: Name of new value column**
**- ...: Names of source columns that contain values**
**- factor_key: Treat the new key column as a factor (instead of character vector)**

benfep_long <- gather(benfep_wide, species, population_density,
uvigerina_peregrina:cibicides_wuellerstorfi, factor_key=TRUE)
```

*benfep_long*

```
**>   station lattitude longitude                species population_density**
**> 1       1    10°56'    178°23'      uvigerina_peregrina              17.3**
**> 2       1    10°56'    178°23'      hoeglundina_elegans            1000.7**
**> 3       1    10°56'    178°23'  cibicides_wuellerstorfi               0.1**
**> 4       2    12°02'    175°59'      uvigerina_peregrina            5273.2**
**> 5       2    12°02'    175°59'      hoeglundina_elegans              50.0**
**> 6       2    12°02'    175°59'  cibicides_wuellerstorfi                42**
```

These are of course only optional suggestions on how the accessibility of the manuscript might be improved. My other point is not really a change in the manuscript itself but more a question/suggestion about the future of this database:

Line 338-342: "This database is conceived as a springboard to store future quantitative data of benthic foraminifera in the East Pacific and make them available to the scientific community. It can be enlarged with new records as they are being generated or after the authors request, therefore providing an ongoing live resource. Any changes to add, correct, or update taxonomic categories to an existing

version will be indicated in PANGAEA. We also encourage users of the BENFEP database to quote original data sources."

How do the authors think about the future of the database? Should there be an easy protocol or "cookbook", how to add data of new records as they are being generated? Are there any permanent members of the group that are able to sustain, clean up and update the database? I think for individual members this might be an impossible task but it should be important to avoid misuse of the feature to update the database. One suggestion would be to provide a review system like in WORMS: Uploaded datasets that are not reviewed, yet, will be marked as "not reviewed". Reviewed datasets might be marked as "accepted" or "unaccepted", if there are any issues considered by a reviewer. Reviewers could be volunteer foraminifera taxonomists. I think there might be a big support in the community for such an effort.

Finally, are there any plans to include other ocean basins into the database in the future? These are just some questions/suggestions about the future of the database and of course, this is an own project by itself and some things cannot be directly integrated into the paper and database. For example, the review system for future datasets would need an own platform or deeper collaboration with Pangaea. Though, I think it is worth it to think about the legacy of such a huge project and maybe to integrate some points about the discussion of the future of the database at the end of the paper about "Data availability and future plans".

---

## Author Comment (AC2)

**Reply to comment on essd-2022-324** (https://doi.org/10.5194/essd-2022-324-RC1) by the referee Lukas Jonkers.

The authors thank Lukas Jonkers for the time on reading and reviewing the manuscript. We sincerely acknowledge all the Reviewer suggestions that aimed to improve the manuscript readability and usability of the BENFEP database.

Our responses (plain text) to point-by-point comments are listed after the Reviewer comments (italic and bold font) followed by the main changes to the manuscript (italic font). Please, refer to the annotated manuscript and companion files.

***Section Introduction: The motivation to focus on the East Pacific remains somewhat unclear. The processes and Variability that the area is exposed to/characterised by (L31-39) do not seem to be unique to the Eastern Pacific. It would hence be good to be clearer about the scientific motivation to build a database for this specific region.***

We agree with the Reviewer´s comment. The Eastern Pacific is unique in that it holds the largest permanent oxygen minimum zone (OMZ) of the oceans. Climate change is currently leading to the expansion of the OMZ which make the Eastern Pacific Ocean especially vulnerable to diversity and habitat loss (as indicated in the references in the text). A synthesis effort to integrate benthic foraminifera geographic distributions would be beneficial for the stakeholders aiming to assess the vulnerability of this ocean area as well as the specialized scientific community aiming to construct modern analogues for the interpretation of the fossil record, in particular those related to paleoO2 reconstruction based on benthic foraminifera.

In the revised version of the manuscript, we have indicated the OMZ as a unique feature of the Eastern Pacific and clarified the scientific motivation to build a database for this region.

Changes to the text: Text rephrased (section 1, Introduction).
*In an historical context, the increase in ocean temperatures and the expansion of the already existing extensive oxygen minimum zone (found about 100 to 900 m water depth, Karstensen et al., 2000) are the major threats to shallow and deep-water benthic ecosystems from the Eastern Pacific (Sweetman et al., 2017; Breitburg et al., 2018; Yasuhara et al., 2019). These attributes make the Eastern Pacific an area of interest for assessing the past, present, and future of the marine ecosystem status and its response to expected environmental changes (e.g., Calderon-Aguilera et al., 2022).*

***Section Introduction: The authors also mention other syntheses. What is the scope of linking them? I understand that this is no trivial exercise, but I think a few lines somewhere (in the outlook) might be beneficial to the manuscript.***

We have added a few lines at the beginning of the penultimate paragraph of the introduction to link another synthesis. There, we highlighted the use of synthesis databases for developing proxies to interpret the fossil record. We also mention the use of those databases to assess the biodiversity change under ongoing climate change and to give the readers some background information about how extensive, both in number of samples and microfossil groups, are the existent databases of planktonic organisms, compared to the benthos.

Changes to the text: Text added (section 1, Introduction).
*The data synthesis of marine microfossils from surface sediments has been a valuable resource among the palaeoceanographic community. They are generally used for constructing modern analogues to interpret the fossil record and, more recently, to evaluate the biodiversity response to ongoing climate change (e.g., Jonkers et al., 2019; Yasuhara et al., 2020).*

*Section Taxonomy: …..Still the data is complex, perfectly illustrated by the fact that the authors recognise 1071 taxa, yet their dataset contains 1502 columns with abundance information. Because of this it would be good to provide a bit more detail on how taxa were mapped onto the WoRMS taxonomy. Was this done using the synonyms provided in WoRMS, using expert knowledge, something else?*

*Whilst looking at the synonym list and the taxon names in the data file I noticed the occurrence of many variants and subspecies, which are treated as separate categories/taxa. This of course risks inflating biodiversity and renders comparison of different studies challenging as subspecies and variants need not be consistently recognised. It would be good if their meaning and how they are treated would be described in the methods of the harmonisation.*

Thanks for your comments. As a requisite for data synthesis is the taxonomic harmonization of the given microfossil group, we decided to strictly follow the taxonomic criteria and synonymy included in WoRMS. We acknowledge that WoRMS is not the only taxonomic possible register for foraminiferal taxa but it is openly accessible to the public and it is continuously updated, which is an asset for a purportedly live database as BENFEP (now renamed BENFEP_v1).

We acknowledge the Reviewer's concern about the incomplete information about the harmonization procedure. Consequently, we have implemented the following changes that we hope to help to bring more transparency to the harmonization procedure:

- We have added a few sentences in section 2.4 to clarify the taxonomic harmonization procedure, as suggested by the Reviewer. There, we explain that we maintained the assignation given in WoRMS, even if the valid name resulted in a variety or subspecies. Additionally, we have increased the taxonomic information of each harmonized species in the Supplement section.

Changes to the text: Text added (new section 2.4, Taxonomic harmonization, previously section 2.5).
*In order to find the valid species name, we searched each author´s original species assignment in the WoRMS research engine. This procedure enables to identify whether the original species name is accepted (valid species) or if it is a synonymous of the valid species or taxa correspond to a variety or a subspecies. When the original species name was not currently in use, it was substituted by the valid species, subspecies, or variety name.*

The number of varieties (14), subspecies (4) in BENFEP_v1 is low compared to the number of total valid taxonomic assignations (1093: including identifications to species and below species level and the two taxa of uncertain status).

Changes to the Supplement: Please, refer to the new Supplement section.
The new Supplement File 1 incorporates more comprehensive taxonomic information (status, rank, order, family, class, AphiaID, etc) about the species harmonized in BENFEP_v1 than the previously submitted file.

Changes to BENFEP_v1 files:
The species numbers we presented in the submitted version have changed in the new reviewed version because we have received additional datasets (Mallon, 2011; Glock et al., 2020; Tetard et al., 2021) and we have included them. Besides, we have added the living counts of Smith (1964, 1973), Patterson et al. (2000), removed some samples with zero total abundance (wrongly included in the previous submission), and we have detected a few errors which are now amended.

*Section Taxonomy: …..In addition, could part of the reason for percentages summing to >100 be related to the reporting of species and subspecies without removing either the subspecies or the species prior to summing? This is a common issue in planktonic foraminifera datasets.*

Thanks for the remark. We reviewed that this is not the case for the species in BENFEP_v1. The percentages summing more and less than 100% were a source of continuous concern during the building up of the database. It is a common feature when data are sourced in percentages. It should be assumed that when the original sources reported the full (all species counted) dataset as percentages, the total sum should be 100%. However, in a few datasets that were digitized or manually digitized (in most of the cases these were reviewed by at least two people) the original datasets do not add to 100%. Therefore, we interpret that the original sources must have errors. In any case, we cannot discard some human errors from our side.

We think that sourcing the data in percentages is prone to errors and precludes some diversity calculations. For this reason, we make a plea to authors to provide raw data in counts (section 4 "Recommendations").

***Section Taxonomy: Some of the taxa appear to be fossil (e.g.https://www.marinespecies.org/aphia.php?p=taxdetails&id=927451), how are these treated? Are they real and therefore indicative of bioturbation/sampling of old material, or do they reflect taxonomic confusion?***

That is right. Indeed, BENFEP_v1 includes 109 taxa identified to species-level considered "fossil only" by WoRMS. The "fossil only" taxa are treated as the considered extant species. We justify our decision with the following arguments:

Firstly, we wanted to provide the users with an unabridged version of the original data. We followed the same criteria for any other non-numerical data originally provided by the authors (section 2.2).
Secondly, there are several reasons why a benthic foraminiferal species taken from a surface sediment sample might appear as fossil: a) It is an actual fossil specimen. In that case, the species might be reworked from nearby older sediments; b) it could represent a mistaken identification in the original source or c) the species is not a "fossil only" species but it has been inaccurately attributed as such by WoRMS. The latter is subject to change as new knowledge emerges; therefore, we consider important to retain the information as it appears in WoRMS. Eventually, this study might lead to revisiting the "fossil range" of some species.

In the new version of the manuscript, we have added the following information that could help the users to find whether a species is extinct or extant (according to WoRMS):

-In new section 2.4 we have now indicated that harmonized species include those considered "fossil only" according to WoRMS.

Changes to the text: Text added (new section 2.4, Taxonomic harmonization, previously section 2.5).
*Some taxa included in BENFEP_v1 are considered as "fossil only" by WoRMS. Nevertheless, we kept those in the database. There are several reasons to explain the occurrence of a species categorized as "fossil only" in a sample; it represents a true displaced fossil species from ancient sediments (reworking), a mistaken identification, and, an extant species inaccurately attributed as "fossil" by WoRMS. As it is not clear which of these circumstances applies in each case, we decided to maintain the species to prevent information losses in case of future re-evaluation of the "fossil range" by WoRMS.*

- In the new section 3.6.4, we have indicated the datasets that specifically mentioned species potentially reworked from old sediments and the relative contribution of those species to the assemblage.

Changes to the text: text added (section 3.6.4, The representativeness of the surface sediment assemblages, previously section 3.6.3).

*One of the purported applications of BENFEP_v1 is to provide a quantitative estimate of recent benthic foraminiferal assemblages that could be later used in palaeoenvironmental interpretations (e.g., Fig. 6). The database integrates quantitative data obtained from oceanic regions with different depositional environments, sedimentation rates, carbonate preservations and types of assemblages, collected over different sampling years and using an array of sampling devices that might result in diversion from recent conditions. For example, dead benthic foraminifera obtained from surface sediments might not be representative of the surface if the sampling device fails to recover the sediment-water interface or sedimentation rates are very low. The 36% of the surface sediment samples were retrieved using different types of coring devices (gravity, piston, dart and Phleger corer, calculations using "dev_1"), which are sampling techniques that can cause perturbation or miss-sampling of the surface sediment (Weaver and Schultheiss, 1990). Since the studies included in our database did not date the surface sediment (except for Palmer et al., 2020), we cannot discard that some samples correspond to pre-Holocene conditions. The most comprehensive compilation of sedimentation rates from core-top samples is from the equatorial Pacific and shows highly variable values, ranging from 0.8 to 14.2 cm/ka (Mekik and Anderson, 2018), meaning that surface sediment samples in this region correspond to recent conditions (assuming that no perturbation occurred during sampling). Reworking, downslope transport and carbonate preservation might be other factors influencing the composition of the assemblages obtained from the surface sediments.*

*The presence of "potentially fossil" species reworked from ancient outcrops (see "remark_2" and Supplement File 4) is included in the datasets of Bandy and Arnal (1957), Echols and Armentrout (1980), Ingle et al. (1980) and Zalesny (1959). Still, they represent less than 5% of the assemblage. The contribution of specimens displaced specimens from shallower locations is also low, as indicated by Bandy and Arnal (1957); Ingle et al. (1980); Harman (1964), Pettit et al. (2013), Uchimura et al. (2017) and Zalesny, 1959). Finally, Pettit et al. (2013) in the Gulf of California, and Boltovskoy and Totah (1987) and Resig (1981) off South America in samples below the carbonate compensation depth, are the only authors mentioning poor preservation of calcareous benthic foraminifera.*

- Because reworking has been mentioned by some authors, we collated the indications of the original authors in "remark_2" column. The "remark_2" of BENFEP_v1 guides the users to look for specific information in the new Supplement File 4.

Changes to the Supplement: Please, refer to the new Supplement section.
The new Supplement File 4 collates the original authors´ indications about potentially reworked species and comments about some potential misidentifications.

-Indications about each species "fossil range" as indicated by WoRMS in the new Supplement File 1

Changes to the Supplement: Please, refer to the new Supplement section.
The new Supplement File 1 includes a column dedicated to the "fossil range" according to WoRMS. We renamed the column as "occurrence" to indicate whether a taxon is extinct or extant.

***Section Taxonomy: The authors have chosen not to preserve the original name and have provided a synonym list in the supplement. The exact changes to each data set are therefore no longer traceable. How do the authors enable/envision future updates to the taxonomy or application of a different taxonomic framework where species lumped here are recognised as separate taxa? And would it be possible to include this information somehow to make the taxonomic framework more flexible and future proof?***

We acknowledged the Reviewer's concern about the incomplete information about the original author's taxonomic concepts procedure which might preclude reusability. In order to make clear to the potential users the taxonomic concepts provided by each author, and bring more transparency to the harmonization procedure, we have now included the following documents in the Supplement section:

Changes to the Supplement: Please, refer to new Supplement section.
-The new Supplement File 2 includes the original author's taxonomic concept for each species harmonized in BENFEP_v1 and included in new Supplement File 1 (please refer to comment above).

-The new Supplement File 3 includes specific remarks on the harmonization procedure.

***Section Taxonomy: One way to address many of these issues is to convert the data to long format. This would allow different taxonomies to exist in parallel (and hence preserve the original). Moreover, by having species, subspecies and varieties in separate columns (perhaps even genus to capture sp/spp) the hierarchy of the taxonomy can be preserved in a way that makes it easier to work with because it facilitates easy grouping at different taxonomic levels. I don't claim that this is the only solution, nor that it is absolutely essential to rework the entire database, but it would be good if the authors reflect more on their methodology and describe how they dealt with these issues or how they would recommend users of the database to deal with them.***

It is our goal to reach all type of potential users interested in benthic foraminifera, regardless of their preferred method to handle large datasets. For those who use more conventional software to read spreadsheet-type files, we keep the previously submitted format (BENFEP_v1_short). Those users will have the original authors' specific assignations in a separate file (Supplement File 2).

For those who prefer to access data using another type of software (e.g., geospatial software), we provide the database in long format (BENFEP_v1_long). The long-format version integrates all the information

provided in the short format plus taxonomic information for each species (File 1), "occurrence" (File 1) and each author´s species assignations (File 2), as suggested by the Reviewer.

Please, refer to section 2.5 of the reviewed version of the manuscript (which integrates former sections 2.4 and 2.6). There we explain the structure of the two formats of the database. Please, see the new Table C1 and Table C2, where we detail the column names and column codes for BENFEP_v1_short and BENFEP_v1_long.

***Section Limitations: There is very little discussion about what "living" actually means or how it is deduced. I think this could be useful for the non-specialist.***
***…. And finally, if living really means living, could the database be used to trace shifts in the assemblages over time given that the database covers a long period of sampling? Would this be a potential use of the data that could be mentioned in the introduction?***

We acknowledge the Reviewer's suggestion. Therefore, in the new version of the manuscript, we have briefly indicated what *living* and *dead* means (section 3.3). We expanded on the issue of using living or dead assemblages for modern analogue construction or to trace changes in the assemblages over time (new section 3.6.4).

Changes to the text: Text added and rephrased (section 3.3).

*The Rose Bengal staining (Walton, 1952) is the only method used by authors to distinguish dead (non-stained) from living (stained) foraminifera at the time of sampling. Living plus dead refers to an assemblage were living (stained) and dead (non-stained) are counted together in the same sample.*

Changes to the text: Text added and re-phrased (new section 3.6.4, previously section 3.6.3).

*BENFEP_v1 includes information of living, dead, and living plus dead assemblages whose suitability for building recent analogues is under discussion among the scientific community. The use of Rose Bengal as "vital" staining could be controversial because attached bacteria or algae, decaying protoplasm of dead individuals might stain, resembling the staining of the protoplasm of a "true" living individual (see review in Schönfeld, 2012). However, it is still the most widely used method to distinguish "living" (stained) from "dead" (non-stained) foraminifera and it is considered reliable if used cautiously. It might be argued that only living foraminifera should be used to consider baseline studies (Schönfeld, 2012). However, it might also be considered that living assemblages represent a "snapshot" of the foraminifera living at the specific time of sampling and do not hold the time-averaged representativeness of the dead assemblages (Murray, 2000). Regarding all these potential concerns, we have incorporated a rich collection of metadata to BENFEP_v1 that can be used by the final users to evaluate data quality and to tailor the final output to their specific criteria.*

***Section Limitations: Conversely, little attention to how old dead could be. Some more explanation and reference to discussion in the literature would be good in this regard. Paragraph 3.6.3 goes some way, but does not cite much literature.***
***The same paragraph also mentions that core top samples may not be representative of the most recent sediment if the sediment water interface is not captured. With the majority of the samples collected using gravity corers this could be a real issue, especially when dead and living specimens are not separated. Some discussion is probably warranted.***

We acknowledge the Reviewer's suggestion. Therefore, we have expanded on the information collated from the publications about age estimates of the surface sediment, foraminiferal preservation, and the presence of reworked specimens in new section 3.6.4 (previously section 3.6.3). Please refer to the reviewed document and to a reply to comment above.

Moreover, BENFEP_v1 contains metadata dedicated to the type of devices ("dev_1, dev_2"), sampling interval ("interval") and type of assemblages ("assemblage") which could aid users in filtering the data following their own criteria.

*Is the preservation potential of all species equal? I imagine that calcareous species are prone to dissolution and that agglutinated species are not always well preserved. Some discussion/guidance on this matter would be good.*

As the Reviewer indicates, a few authors of the datasets included in BENFEP_v1 mentioned poor preservation of calcareous forms: Pettit et al. (2013) in the Gulf of California and Resig (1981), Boltovskoy and Totah (1987) in samples located at water depths deeper than 3700-4100 m. Indeed, BENFEP_v1 incorporates datasets from the lower abyssal zone and below the carbonate compensation depth in the Eastern Pacific which are dominated by species with non-calcareous walls (Smith, 1973, Resig, 1981; Burmistrova et al., 2007; Enge et al., 2012).

We have now mentioned the authors indicating deficient carbonate preservation in new section 3.6.4 (previously section 3.6.3). Please, see reply to comment above

Regarding the preservation potential of foraminiferal species, benthic foraminifera are less susceptible to dissolution than planktic foraminifera and dissolution susceptibility varies among taxa. However, with 1093 valid taxonomic designations, the ranking of taxa solubility falls beyond the scope of this study.

*Section Limitations: Usage notes*
*The database is very heterogeneous. There are differences in the sampling instruments, sampling depths, picking method and in the level of detail in the taxonomy (some studies appear to report subspecies and variants, others don't) and the data is sometimes non numeric.*
*This is of course not the authors' fault, but it poses some important limitations on/challenges for the use of the data. I therefore miss some usage notes or recommendations on how to deal with this heterogeneity.*

*For instance:*
*How to handle non-numeric data;*
*How compare samples analysed using different size fractions;*
*What is the meaning of empty cells;*
*What to do when assemblages don't sum to 100%;*
*How to derive meaningful indices of biodiversity given the differences in the level of taxonomic detail; etc.*
*Some of these questions are perhaps trivial to the authors, but not all users of the data will be benthic foraminifera specialists and they might benefit from more detailed user instructions*.

We acknowledge the Reviewer's suggestions, and willing to help the final users, we have added the following changes to the main text and files. Please, find explanations to the Reviewer´s comments below:

***Non-numerical data:*** The handling of non-numerical data depends on the user's final purposes. For example, if an user wants to calculate sample species richness or species presence, they could replace the non-numerical data with 1, as the non-numerical data mostly indicates the presence of a species. In the case of using the data for percentage calculations, we would suggest removing them as they cannot be translated into a particular numerical value.

We have added a new section (3.6.3) providing general guidance regarding non-numerical data.

Changes to the text: Text added (new section 3.6.3. Non-numerical data).
*There are 18 datasets which include non-numerical data ("x", "<1") in their records (see "remark_1"). Those data might interfere in the calculation of the relative abundances and some diversity indexes (e.g., Shannon Weaver). However, they provide useful information on species presence and therefore they are potentially useful for biogeography and calculations of species richness. General suggestions on how to manage non-numerical data in R can be found in Supplement File 5.*

Changes to the Supplement: Please, refer to the new Supplement section.

-The new Supplement File 5 includes some suggestions on how to load the short format of the database in R and perform basic operations (e.g., build up a percent file), and manage non-numerical data

***Empty cells:*** this was indicated in the former Table B1: "An unfilled field, indicates no information available. It applies to any column". We have added the meaning of an empty cell in new Appendix C section (Table C1 and Table C2) and we have indicated the meaning of an empty cell in the new reviewed version of the manuscript (section 2.5): "An empty cell in any column indicates that there is no information available"

Changes to the Appendix: Please, refer to the new Appendices section.
Table C1 and Table C2 (previous Table B1) have been re-done to include explanatory notes on column names and column codes of BENFEP_v1_short (Table C1) and BENFEP_v1_long (Table C2).

***What to do when assemblages don't sum to 100%:***

We do not have a straightforward answer for that. Unfortunately, it is a common pattern in samples sourced from percentage. Our preferred choice is keeping the original data, rather than recalculating each individual percentage to 100%. This specially applies when data come from assemblages where authors do not provide the full scope of species or those which contain non-numerical data. As we make the user aware of this situation (section 3.6.2 and "remarks_1", "remarks_3" of the BENFEP_v1 files), we leave up to the final user the choice of excluding these samples from further analyses, recalculating the abundances, or keeping the original abundance data.

***How compare samples analysed using different size fractions;***
***How to derive meaningful indices of biodiversity given the differences in the level of taxonomic detail;***
***etc.***

We consider that handling species and subspecies and size fraction is the users´ choice. They might choose adding a subspecies to the parent species and select samples according to a specific size range. The number of taxa below the species level ("varieties" and "subspecies") is low (18) compared to the total number of taxa to species level (1073 plus 2 taxa of uncertain status). See new Supplement File 1.

***Data section_ The format (xlsx) is proprietary and thus therefore not strictly adhering to the FAIR data standards. I recommend making the data available as some kind of text file that is universally readable and more likely to remain so.***

Being in completely agreement with the Reviewer, and in order to support the reusability of digital assets, we now provide BENFEP_v1_short and BENFEP_v1_long as text files.

***Data section_ The species names in the headers appear inconsistently formatted and not to map one-to one to the names in the synonym list in the supplement (use of underscores and brackets). I realise that this is a pain, but it would enhance machine readability if the formatting were made consistent.***

We understand and agree that consistent formatting is a must. Consequently, we have checked that the headings of BENFEP_v1_short do match the headings of the species list in the Supplement (File 1, File 2).

***Data section_Another, very small remark, is the use of special characters in the column headers, which complicates (machine) reading and the authors could consider replacing them with plain text.***

We understand and agree that the use of special characters in the column headers complicates machine reading. Accordingly, we have made the following changes in BENFEP_v1 to facilitate usability. The changes are explained below:

Changes to BENFEP_v1 files:
-simplification of column headers: taxonomic designations are simplified to contain only the genus and species name (or varieties, subspecies) in plain text. Thus, we have removed information of authority, year

or any special character (e.g., parenthesis, &, etc). The users could find this information in Supplement File 1 or in BENFEP_v1_long in columns "valid_authority".

-removal of capital letters from column names and restrict column length to 10 characters to facilitate use in GIS software.

-use of spaces: albeit some exceptions (taxa names), we try to avoid using spaces between words in cells and column names. In case a space is required, we added an underscore.

***Minor comments***
***L10: consider adding "benthic" at the start of the sentence.***

Done as suggested

***L29: I don't understand what "figures" means here?***

We have re-phrased the text in that line to:

*Several areas of Eastern Pacific Ocean are at severe risk of species loss (Finnegan et al., 2015; Yasuhara et al., 2020, UNESCO, 2022) and consequently, some of them have been categorized as marine protected areas (Enright et al., 2021).*

***L58: reword, something like "rendering interpretation of the fossil record more meaningful."***

Done as suggested

***L171-172: I assume that the database can be handled in any software, especially once converted to text format. Why focus on these two examples only instead of highlighting that it can be analysed using any software on any platform?***

Thank you for the suggestion.
Because we provide BENFEP_v1 in a text file (which is a more universal and license-free format) we followed the Reviewer's suggestion and added the text accordingly.

Changes to the text: Text rephrased (new section 2.5 Structure of the database, previously section 2.6)
*The database in its two versions is presented in text format and can be managed with virtually any software.*

***Fig. 2 legend: the colour scale does not, I presume, show the log10(number of species). I think it shows the number of species as normal numbers, but with a logarithmic colour scaling. I would just remove the (log10).***

The reviewer is right. We have modified the figure in the reviewed version of the manuscript.

Changes to the Figures:
Figure 2 has been re-made to amend the legend and to update the changes made in the files (see previous explanation to Changes to BENFEP_v1 files).

***Fig. 4: perhaps this is a figure where I wonder what the exact purpose is. Do we learn much about the data and why is that useful?***

Changes to the Figures:
We have removed the former Figure 4 from the reviewed version of the manuscript.

***L223: these size ranges seem inconsistent with the figure below, those shown in the figure are wider. Why? And are the ">" signs needed?***

We are sorry that the text and the figure were not clear enough. We now provide an alternative version of the figure and rephrased the text accordingly. We believe this new plot is clearer than the former one.

Changes to the text: Text added and rephrased (section 3.3).
*Most of benthic foraminiferal assemblages were analyzed in the smallest size fraction commonly used in benthic foraminiferal studies. For example, 65.4% of the samples were analyzed using 42, 61, 62, 63 and 74 μm as the lower end of size fraction (e.g. assemblages where studied in the >42 μm, >61μm, >62 μm size fraction, etc.). The 6.1% is using 88 and 105μm and 17.7% the 125, 149, 150, 200, 212 and 500 μm as the lower end of size fraction.*

Changes to the Figures: new Figure 4 (previously Figure 5).
In the reviewed version of the manuscript, we have re-done Figure 4. We have calculated the relative contribution of each size fraction (>61, >62, etc) independently and represented it as a portion of the pie chart in the new Figure 4.

***Fig 7D: are there really samples with 0 species? And how exactly was this figure made, see my comments above on user instructions.***

This figure was built up by calculating the number of taxa per sample. In a few sampling points there were no foraminifera (0 abundance and 0 number of species). It was inconsistent to include those samples in the previously submitted version. Therefore, we have now removed those samples from the new files of the BENFEP_v1 database.

Changes to the Figures:
In the new reviewed version, we have re-plotted Figure 6D (previosuly Fig. 7D) with the information of the new BENFEP_v1 files.

***Fig. 7E: I am not sure in which direction the abundances are summed in this figure (rows, columns or both?). If the authors want to show the depth distribution of selected species then I would expect that each row should sum to 100%, but that seems not the case.***

The figure was built up by calculating the mean relative abundance of each of the selected species for each bathymetric range. We have now indicated that in the figure caption *(....and selected species mean relative abundance with water depth (E) in BENFEP_v1)*

Changes to the Figures:
In the new reviewed version, we have re-done Figure 6E adjusting it to the bathymetric ranges of Morkhoven et al., (1986). The bathymetric ranges we used in the submitted version were incorrect as they were designed for the pelagic environment.

***L287: "A few samples…" I counted 423 and 98 >105%, that is not a few. Just provide the numbers.***

Thanks for the annotation. This sentence was re-written to "well above 100%" and we indicated the entry it refers specifically to (Butcher, 1951). As we mentioned earlier, not summing 100% is a common feature of the data sourced in percentages.

***L331-332: something went wrong with the sentence "It is recommended It would be…"***

Thanks for the annotation. The sentence has been amended.

*L333-334: the authors should be bolder here, providing taxonomic information is crucial for the reusability of (any) taxonomic data. So it is not only of value to the "specialised foraminifera community" and unclear taxonomic information compromises the data and prevents their reusability. Using data only once (in one study) really is a waste of time!*

We have followed the Reviewer's recommendation and we have changed the text accordingly.

Changes to the text: Text rephrased (section 4 Recommendations, previously section 5).
*Taxonomic information and supporting images are crucial elements for reliable taxonomic harmonization and data reusability.*

*L340: I recommend some kind of versioning scheme so updates can be more easily recognised.*

The reviewer is right; we did not consider versioning properly.

We renamed the database file to BENFEP_v1 to differentiate it from future versions.
Section 5 contains now information about how we foresee versioning after BENFEP_v1.

Changes to the text: Text added and rephrased (new section 5. Data availability and future plans, previously section 6).
*The BENFEP_v1 database can be accessed from https://doi.org/10.1594/PANGAEA.947086 (Diz et al., 2022b). This database is conceived as a springboard to store future quantitative data of benthic foraminifera in the East Pacific and make them available to the scientific community. It will be open for any new quantitative data entry and thus, it welcomes any new data published or provided by any contributor. The database will be updated by the authors once a considerable number of new entries need to be incorporated or changes are required to update taxonomic categories to an existing version. New versions of BENFEP will be submitted and curated in PANGAEA. Collaborations with individual researchers and institutions are welcomed specially regarding potential expansion to other ocean basins.*

---

## Author Comment (AC3)

**Reply to comment on essd-2022-324 (https://doi.org/10.5194/essd-2022-324-RC2) by an Anonymous Reviewer.**

The authors thank the Anonymous Reviewer for the time on reading and reviewing the manuscript. We sincerely acknowledge all the Reviewer suggestions that aimed to improve the manuscript readability and usability of the BENFEP database.

Our responses (plain text) to point-by-point comments are listed after Reviewer comments (italic and bold font) followed by the main changes to the manuscript (italic font). Please, refer to the annotated manuscript and companion files.

***Comments on data usability: Supplement 1 indicates which names used by the original authors were synonymized with a given taxonomic name from WORMS, however it does not indicate which authors (or publications) used which names. If future taxonomic revisions are made such that previously synonymized names are assigned differently, there would be no way to deconvolve the data.***

We acknowledge the Reviewer's concern about the incomplete information about the harmonization procedure which might preclude reusability. In order to make clear to the potential users the taxonomic concepts provided by each author, and to bring more transparency to the harmonization process, we have now included the following documents in the Supplement section:

Changes to the Supplement: Please, refer to the new Supplement section.

-The new Supplement File 1 incorporates more comprehensive taxonomic information (status, rank, order, family, class, AphiaID, etc) about the species harmonized in BENFEP (now BENFEP_v1) than the previously submitted file.

-The new Supplement File 2 includes the original authors' taxonomic concept for each species harmonized in BENFEP_v1 and listed in new Supplement File 1.

-The new Supplement File 3 includes specific remarks on the harmonization process.

***Comments on data usability: If the data were provided as long data, then each entry could have both the synonymized name from WORMS and the name from the original authors. It would also be clear where subspecies were consolidated if other researchers wish to consider that aspect of diversity.***
***Because the data are sparse (have many zeros), using a long data format would also reduce the file size (not that it is very large as is). Further, adding new data to the database would be substantially easier because it could just be appended to the end of the existing data without having to add or modify columns. Using long data would also allow new data to be uploaded as separate compatible files without modifying the original contribution and allow other workers to more easily use the resource.***

Thanks to the Reviewer for the suggestion, which has been also made by other two Reviewers. It is our goal to reach all type of potential users interested in benthic foraminifera, regardless of their preferred method to handle large datasets. For those who use more conventional software to read spreadsheet-type files, we keep the previously submitted format (BENFEP_v1_short). Those users will have the original authors' specific assignations in a separate file (Supplement File 2).

For those who prefer to access data using another type of software (e.g., geospatial software), we will provide the database in long format (BENFEP_v1_long). The long-format version integrates all the information provided in the short format plus taxonomic information for each species (File 1) and each author´s species assignations (File 2).

Please, refer to section 2.5 of the reviewed version of the manuscript, which integrates former sections 2.4 and 2.6. There, we explain the structure of the two formats of the database. Please, see the new Table C1 and Table C2, where we explain column names and column codes for BENFEP_v1_short and BENFEP_v1_long.

*The authors note that R could be used to transform the data such that it is usable in ArcGIS, but do not recommend a function nor describe issues the user may encounter when doing so and how to avoid those issues. It would, of course, be better to provide the data in long format and then tell the reader how to turn it into wide data for ecological analyses in R. It would also be helpful to tell readers how to amend the formatting of the provided data table so that is it is readable in R. As presently formatted, a user could not simply read it into R.*

We agree that it would be useful for readers to have some guidance on how to load BENFEP_v1_short in R. Consequently, we have now included a new file in the Supplement section dedicated to those explanations.

Changes to the Supplement: Please, refer to the new Supplement section.
Supplement File 5 now includes some suggestions on how to load the short format of the database in R, perform basic operations (e.g., build up a percent file), and to manage non-numerical data.

We would like to indicate that the previously submitted version of BENFEP provided as .xlsx format could be loaded in R. All submitted figures (except for the figures made in ArcGIS, and former Figure 5 which used an unabridged version of the previous Supplement) were made by sub-setting and managing the database directly from R.

*Comments on data usability: The authors note that the database will be continually updated. What will be the mechanism of update? Will the Pangea file be replaced or will multiple versions be available through the same Pangea number? Is there a mechanism that will keep the database location stable? How will different versions be annotated such that version can be tracked? If errors are found and corrected in existing data, how will these changes be documented?*

The reviewer is right, and we regret not having delved further into the process of updating the database. In order to facilitate updating, we have re-named BENFEP as BENFEP_v1 along the manuscript and in the PANGAEA repository. Besides, we have now clarified in the reviewed version of the manuscript how we plan to update BENFEP_v1 in the future.

Changes to the text: Text added and rephrased (section 5. Data availability and future plans, previously section 6).
*The BENFEP_v1 database can be accessed from https://doi.org/10.1594/PANGAEA.947086 (Diz et al., 2022b). This database is conceived as a springboard to store future quantitative data of benthic foraminifera in the East Pacific and make them available to the scientific community. It will be open for any new quantitative data entry and thus, it welcomes any new data published or provided by any contributor. The database will be updated by the authors once a considerable number of new entries need to be incorporated or changes are required to update taxonomic categories to an existing version. New versions of BENFEP will be submitted and curated in PANGAEA. Collaborations with individual researchers and institutions are welcomed specially regarding potential expansion to other ocean basins.*

We have consulted PANGAEA and they informed us that the new versions will have a different DOI identifier. However, there is the possibility to link newer and older versions and incorporate information about the new changes included in each version. Likewise, newer versions of BENFEP will be identified with the corresponding version number (currently v1).

*Comments on data usability: Guidance on how to handle non-numeric data would also be warranted. What are the recommended process for including or excluding these entries? Would the recommendations be different depending on the situation or scientific questions? I would recommend using different non-numeric symbols for the different meanings. For example, if x does not mean <1% in all cases a user could not treat all x's in the same way and would have to carefully determine which x's mean what. In some cases, the meaning of the x does not appear to be described in the Notes column and/or is not consistently described in the same Remarks column such that automating that process would be difficult. For example, Bandy_Arnal_1957 has x's, but no indication as to what they mean in the Remarks. For Smith_1964 the Remark is in Remarks 2 rather than Remarks 1 like most of the remarks concerning x's.*

We agree with the reviewer in that non-numerical data influence the management of the database. In any case, handling non-numerical data depends on the user's final purposes. For example, if an user wants to calculate sample species richness or species presence, they could replace the non-numerical data with "1", as the non-numerical data mostly indicates the presence of a species. In the case of using the data for percentage calculations, we would suggest removing them as they cannot be translated into a particular numerical value.

In order to guide users on the management of non-numerical data, we have now added a new section (3.6.3) dedicated to providing some explanations and referring the users to File 5 where they could find more practical information.

Changes to the text: Text added (new section 3.6.3. Non-numerical data).
*There are 18 datasets which include non-numerical data ("x", "<1") in their records (see "remark_1"). Those data might interfere in the calculation of the relative abundances and some diversity indexes (e.g., Shannon Weaver). However, they provide useful information on species presence and therefore they are potentially useful for biogeography and calculations of species richness. General suggestions on how to manage non-numerical data in R can be found in Supplement File 5.*

We inadvertently omitted to indicate the meaning of non-numerical data of Bandy and Arnal, 1957. This is now amended in the reviewed files.

The Reviewer is right in that the non-numerical data ("x", "<1") do not have the same exact meaning in each dataset but they could be summarized in four categories: species representing less than 1%, species absent from the split but present in the sample, fragments, and unknown. We considered that coding the different meanings of non-numerical data would result in confusion. In all the studies, except for Uchio (1960), "remark_1" explains the meaning of the symbol in the original publication (for Uchio dataset we could not find the meaning of "x"). Therefore, our suggestion is to use symbols as species "presence", but exclude them for analyses that require quantitative data.

We acknowledge the Reviewer's annotation about the remark section. Consequently, we have re-arranged the "remarks" columns in BENFEP_v1 sorting them by categories and adding this information to the text and Appendices accordingly. Please, refer to Tables C1 and C2.

Changes to the text: Text added (section 2.5, Structure of the database, integrates previous sections 2.4 and 2.6).
*Meaningful annotations regarding the sample entry was spared in seven columns dedicated to the meaning of non-numerical data, comments about some species, assemblage characteristics, volume of the sample (when data are provided in density), size fraction, sample geolocation and others.*

Changes to the Appendix: Please, refer to the new Appendices section.
Table C1 and Table C2 (previous Table B1) have been re-done to include explanatory notes on column names and column codes of BENFEP_v1_short (Table C1) and BENFEP_v1_long (Table C2).

***Comments on data usability: Given the data are provided in different size fractions, some guidance on whether these data should be analyzed together or if they must be analyzed separately would be warranted. How should users handle sources that record only calcareous taxa? Is the scope of the data source clearly noted in the metadata and connected to the main data file such that a user could easily subset data with different characteristics? Information is provided in the remarks columns, but not in a standardized way that would allow easy subsetting.***

Thank you for the annotation, but we cannot provide guidance on how to analyse data from different size fractions or type of test because we do not know what final user's goal would be when using BENFEP_v1. The metadata of BENFEP, specifically columns named "assemblage", "format", "fraction", "n100", "n200", "n300" were specifically designed and coded accordingly for filtering the samples; per size fraction, the type of assemblage, the format of the data, or the number of specimens analysed in a sample (see Table C1 and Table C2). This provides a ready-to-use method to filter different size fractions.

We have tested filtering and data assembling with "tidyverse" in R. Figures presented in the former submission and in the reviewed version of the manuscript are made this way. The "remarks" section is not

intended to provide this kind of information but only meaningful data about the meaning of symbols, geolocation, size fraction issues, etc. Please, refer to the reply above regarding the columns dedicated to remarks.

Regarding the calcareous and non-calcareous foraminifera. In BENFEP_v1_short, the foraminiferal taxa are not ordered alphabetically. The genera and species with non-calcareous tests are placed from column 24 to column 527. After the column "Indeterminate agglutinated", it follows genera with calcareous test (except for some species of the order miliolina with agglutinated walls). Consequently, in the case of short format finding non-calcareous and calcareous taxa is nearly straightforward. In BENFEP_v1_long, users should look for the classes, families, and genus with calcareous/non-calcareous tests.

***Comments on data usability: In the BENFEPfile1 with the main data, sometimes there is a "0" in a cell but often the cell is empty. Is there a different meaning for a "0" as for an empty cell? When converting from wide to long data, entries with a "0" would be retained and need to be removed by the user unlike the entries where the cell is empty. Formatting all data consistently would be a best practice.***

Thanks for the comment.
The "0" were slips in two entries: Butcher (1951) and Belanger et al. (2016). They should have been empty cells. We have now removed the "0.0" value from the species abundance data of these two entries.

***Line 168: I'm not clear what is meant by "ranked abundance of individuals" and the N100, N200, N300 categories based on the text description. Looking at the spreadsheet, it seems this designates whether a sample had at least 100, 200, or 300 individuals in the total sample. Ensuring the language used in the text is the same as what is used in the spreadsheets would reduce any confusion.***

We agree that the description was not clear. In the new version of the manuscript, we have re-phrased that sentence to match the information in Tables C1 and C2 (previously Table B1).

Changes to the text: Text rephrased (section 2.5, structure of the database, integrates previous sections 2.4 and 2.6).
*Additionally, in columns 1556-1565 we coded whether the number of counted individuals in each sample is equal to or higher than 100, 200 and 300 individuals.*

***Line 159: If the original authors of the data sets do not note whether a specimen is calcareous or not, placing it under "Indeterminate calcareous" may be in error. An "Indeterminate unknown" category would be more accurate.***

We agree with that, and therefore we have created a new column in BENFEP_v1 to include all indeterminate unknown foraminifera "Indeterminate unknown". This column in BENFEP_v1_short is the last one including quantitative data (column 1554) just before column ("total").

***This is also another example of where the language of text does not match the language of the spreadsheet; in the spreadsheet there is a column "Calcareous Indeterminated" but no "Indeterminate calcareous."***

Last minute language revision of the manuscript made us change to "Indeterminate calcareous" while "Calcareous Indeterminated" was already curated in PANGAEA. We have now amended that in the BENFEP_v1 files changing column names to "Indeterminate calcareous" and "Indeterminate agglutinated".

***The column labeled "total" is confusing as there is not an accompanying column for the units of that total. Units of density may vary and it would be useful to know the actual units rather than just "counts per volume unit."***

The column total should be used in conjunction with the format, otherwise, it leads to confusion. This was indicated in the main text, former section 2.6, it read: "column: Total, see also column: Format". We have re-phrased the text to better clarify it in the reviewed version of the manuscript.

We considered that splitting the column dedicated to the sum of species counts per sample into 3 "total" columns: "total_counts", "total_percent", "total_density" could be even more confusing. As mentioned above, metadata enables filtering the database by "format".

Changes to the text: Text rephrased (section 2.5, structure of the database, integrates previous sections 2.4 and 2.6).
*A column representing the sum of species abundance per each sample (column "total") was added at the end of the species quantitative data. Users should check the column "format" for indications whether the value in the column represents the sum of percentages, counts or densities.*

Following the Reviewer's suggestion, we have now added the units when data are provided in density. This information is placed in the dedicated column (remark_4). Please, refer to the comment above.

***Line 104: I'm not clear what the authors consider to be quantitative vs. qualitative data. Does quantitative include raw counts, relative abundances, and densities whereas qualitative is just occurrence? Or does qualitative means something else? Perhaps just stating the type of data would alleviate confusion.***

We are sorry for the confusion created. We have now better clarified what we consider quantitative at the beginning of section 2.2 (Data source and selection protocols)

Changes to the text: Text added (section 2.2).
*We consider data as quantitative when the species abundance in an assemblage is provided as number of individuals (counts), relative abundance (percent) or density (number of individuals per volume unit).*

***The BENEPqual datafile lists localities with metadata and notes the type of data it provides, often noting that the data are available in or are semi-quantitative or are given a species groups. I am not clear what the authors feel a user should do with this information and some description of how this table could be used would be helpful. If these are localities where at least species lists could be obtained, providing the occurrence data rather than just the metadata would make this table of some use for diversity studies.***

It was unfortunate the way we named the database BENFEPqual and the way we described its content because resulted in confusion.

"BENFEPqual" contained geolocations of samples which carried out benthic foraminifera assemblage studies but whose data were not available in the publications but presented in graphs, as non-quantitative information (rare, dominant, abundant) or species presence. The application of "BENFEPqual" is to geolocate samples/studies in the Eastern Pacific, identify the author, and the type of data and refer to the original source for species information.

Because this dataset is creating confusion, we re-organized the text referring to it as follows:

-We now refer to the dataset as a collection of geolocations of studies that could not be included in BENFEP_v1 because they do not meet the criteria for quantitative data. We provide access to the file in the main text and in the caption of Figure B1.

Changes to the text: Text added (section 2.2)
*There are 31 documents published between 1929 and 2019 characterizing assemblages of living and dead assemblages of benthic foraminifera from surface sediments in the Eastern Pacific that could not be incorporated in BENFEP_v1 because species assemblage data are provided in graphs, as species presence or range of abundances (e.g. common, rare, abundant). The geolocation of the samples and the authors of those publications can be accessed from https://doi.org/10.1594/PANGAEA.947114 (Diz et al., 2022a) and they are represented in Figure B1.*

*Figure B1. Spatial distribution of samples in the Eastern Pacific from studies which do not provide quantitative assemblage data. The numbers refer to each author´s dataset. Sample geolocation and metadata can be found in https://doi.org/10.1594/PANGAEA.947114 (Diz et al., 2022a). The procedure for stations´ georeferencing and column coding follows the indications of sections 2.3 and 2.5. The map was made using ArcGIS software version 10.8.2. The global relief model integrates land topography and ocean bathymetry (Sources: Esri, Garmin, GEBCO, NOAA NGDC, and other contributions).*

-We have removed references to the database by the name of "BENFEPqual" along the text. Consequently, we have eliminated previous section 4 "Complementary information to BENFEP: BENFEPqual".

We considered that these changes will contribute to alleviate confusion and improve the readability of the text.

***I notice that Culver and Buzas's Smithsonian Contributions are included in the reference list, however not all the papers they drew from are present in BENEP or BENEPqual. What was the criteria by which sources in those compilations were excluded? Although the authors describe the method by which they search for the data, the criteria by which found sources were rejected is not described and need to be for a user to understand which scientific questions the data are suitable for answering.***

As we indicated in section 2.2, the compilations of Culver and Buzas (1985, 1986, 1987) were an invaluable source of information to locate original benthic foraminifera quantitative studies. However, not all original studies (listed below) included in the three Culver and Buzas´ compilations could be integrated into BENFEP_v1, because they lacked quantitative data. Please, refer to the comment above about the definition of quantitative data we used in this study.

Considering that criteria, we include in BENFEP_v1 the metadata, geolocation and the species abundance of the studies of Bandy and Arnal (1957), Brenner (1962), Butchner (1951), Berger and O´Neil (1979), Erskian and Lipps (1977), Harman (1964), Landkford and Phleger (1973), McGlasson (1959), Morin (1971), Phleger (1964, 1965), Scott (1976), Smith (1973), Uchio (1960), Walton (1955), and Zalesny (1959).

We exclude from BENFEP_v1 the following two groups of publications:

The first group, incorporate studies for which quantitative assemblage data were not available in the original sources. These studies are: Arnal et al., (1980), Bandy (1953, 1961, 1963), Bandy et al. (1964a, b, c, 1965a), Cockbain (1963), Cooper (1961), Jones and Ross (1979), Martin (1930, thesis), Moyer (1929), Reiter (1959), Resig (1958, 1960), Todd and Low (1967), Watkins (1961). The geolocations of those studies are now accessed through the link indicated in the caption of figure B1 (https://doi.org/10.1594/PANGAEA.947114) .

The second group integrates taxonomic studies dealing exclusively with particular species occurrences (not assemblage-integrated data): Angel (1975), Arnal (1955), Bradshaw (1961, 1968), Bush (1930), Cushman (1910, 1913, 1915, 1917, 1925, 1927), Cushman and Grant (1927), Cushman and Martin (1935), Cushman and McCulloch (1939, 1940, 1948, 1950), Cushman and Moyer (1930), Cushman and Todd (1943, 1947), Cushman and Valentine (1930), Church (1929a,b, 1954, 1968), Detling (1958), Hanna and Crouch (1927), Douglas (1964), Goes (1896), Hamlin (1960), Lalicker and McCulloch (1940),  Harrington (1956),  Maurer (1968), McDonald and Diediker (1930), Nicol (1944), Palmer (1929), Rankin (1931), Scott (1976a, b), Sliter (1968, 1969, 1970, 1970), Phleger and Soutar (1973), Whiteaves (1886).

Finally, we could not have access to the following studies referenced in Culver and Buzas (1985, 1986, 1987): Bandy et al., 1965b, Echols (1969); Martin (1931, 1936), Natland (1933, 1938), Phleger (1951, 1957, 1967), Resig (1965), Steinker (1976), Scott (1974).

In this study, we prioritize integrating original sources which quantitative data, therefore merging Culver and Buzas (1985, 1986, 1987) datasets into BENFEP_v1_long would result in duplicated records.

*If the original authors collected surface sediment samples and simply did not stain the sample to determine what is living, would that go under "living and dead" or "dead"? It seems that "dead" should only be applied to samples where staining was done and the stained and unstained individuals were counted separately. If the original authors simply did not do staining a surface sample will usually contain both living and dead individuals and I believe it is typical to consider this samples as "living and dead".*

We appreciate the annotation of the Reviewer. "living and dead (LD)" referred only to studies that specifically stated that they studied living (rose bengal stained) and dead (un-stained) foraminifera from the same sample. In order to avoid confusion, we have changed living and dead to "living plus dead" along the text, Figures and in the Video supplement.

We have now explained what means living, dead, and living plus dead in the reviewed version of the manuscript section 3.3.

Changes to the text: Text added (section 3.3).
*The Rose Bengal staining (Walton, 1952) is the only method used by authors to distinguish dead (non-stained) from living (stained) foraminifera at the time of sampling. Living plus dead refers to an assemblage were living (stained) and dead (non-stained) are counted together in the same sample.*

We would like to explain that we did not artificially sum live and dead counts and when we curated "LD" is because the authors provided the living and dead counts combined.

However, in three entries we did not include the Living assemblage which the author provided separately. This happened in Smith (1964, 1973) and Patterson et al., (2000). We have now included these living samples in the reviewed version of BENFEP_v1.

*In the case of Morin 1971, for example, both living and dead data are given for the same number of samples suggesting the samples were examined for both and they were counted independently, but in Niensted, 1986 only "dead" is noted even though the sample is 0-2 cm and presumably would contain living individuals, just none were confirmed living.*

That is correct, Morin (1971) provides independent counts for live and dead assemblages, and this is reflected in the database and in Table A1. As indicated before, we did not sum Live and Dead counts when the authors provided them separately.

Nienstedt (1986) did not stain the samples and his/her assemblages are curated as dead (un-stained) ones.

*Some description in the text as to what "living" and what "dead" mean in the text would be warranted and the data attributes made to match how benthic foram workers typically use the terms.*

Please, refer to comment above.

*The authors include information on the picking method and it would add clarity if they also described how different picking methods affect the species that are found such that a user may decide how to subset their data.*

Regarding this issue, we consider that the implications of different picking methods for the species assemblage composition are something that is discussed in the literature. BENFEP_v1 database includes a wide array of techniques which have changed over the last seven decades. We consider that deciding the strengths, weaknesses or suitability of each method depends on the final purpose of the BENFEP_v1 database. We, therefore, prefer not to give any opinion on the different picking methods, and we refer the readers to the review by Schönfeld (2012), cited in our manuscript, for addressing the methodologies on recent benthic foraminifera studies.

***Why is it important that the preparation methods changed over time (Figure 4)? How does that limit what can be done with the data? Does N/A in Figure 4 actually mean "not given" rather than "not applicable"?***

Changes to the Figures:
We have removed previous Figure 4 from the reviewed version of the manuscript.

***Similarly, how does the collection device potentially affect the data or is this not relevant to how a user might subset the data (Figure 3)?***

The device used to obtain the sample might be relevant to evaluate the representativeness of the surface sediment assemblages as recent analogues. The discussion about this topic is now expanded in the reviewed version of the manuscript (section 3.6.4)

Changes to the text: Text added (new section 3.6.4, previously section 3.6.3).
*One of the purported applications of BENFEP_v1 is to provide a quantitative estimate of recent benthic foraminiferal assemblages that could be later used in palaeoenvironmental interpretations (e.g., Fig. 6). The database integrates quantitative data obtained from oceanic regions with different depositional environments, sedimentation rates, carbonate preservations and types of assemblages, collected over different sampling years and using an array of sampling devices that might result in diversion from recent conditions. For example, dead benthic foraminifera obtained from surface sediments might not be representative of the surface if the sampling device fails to recover the sediment-water interface or sedimentation rates are very low. The 36% of the surface sediment samples were retrieved using different types of coring devices (gravity, piston, dart and Phleger corer, calculations using "dev_1"), which are sampling techniques that can cause perturbation or miss-sampling of the surface sediment (Weaver and Schultheiss, 1990). Since the studies included in our database did not date the surface sediment (except for Palmer et al., 2020), we cannot discard that some samples correspond to pre-Holocene conditions. The most comprehensive compilation of sedimentation rates from core-top samples is from the equatorial Pacific and shows highly variable values, ranging from 0.8 to 14.2 cm/ka (Mekik and Anderson, 2018), meaning that surface sediment samples in this region correspond to recent conditions (assuming that no perturbation occurred during sampling).*

Micropalaeontology community is familiar with the benefits and caveats of using a particular sampling device for obtaining surface sediment samples, as these have been discussed in the literature (e.g. Schönfeld, 2012). It is left to the users to evaluate data quality following their own criteria and using the metadata provided in BENFEP_v1.

***How does the distribution of living and dead sampling in the database along latitude impact the analyses that can be done (Figure 2)? While these facts about the data are useful, discussion of how to use the facts (beyond it showing where more data should be collected) would greatly help a reader.***

BENFEP_v1 provides the user with the tools to filter data according to their own criteria and to decide whether data are robust or meaningful for their own purposes.

*Minor notes:*
*Line 22: Last sentence of abstract seems to be missing something and could use revision for clarity ("studies dearth of quantitative data" … do you mean "with quantitative data" or "without quantitative data")*

We have removed this sentence from the abstract.  Please, refer to the comment above where we explain the re-organization regarding the formerly named "BENFEPqual" database.

***Line 26: Please cite the source for the Large Maine Ecosystem scheme.***

In the reviewed version of the manuscript, we have now added Sherman (1991) as a reference for the LME.

***Line 29: What are "protection figures"? Please revise for clarity.***

We have re-phrased the text in that line to:
*Several areas of Eastern Pacific Ocean are at severe risk of species loss (Finnegan et al., 2015; Yasuhara et al., 2020, UNESCO, 2022) and consequently, some of them have been categorized as marine protected areas (Enright et al., 2021).*

***Line 102 says "391 benthic foraminiferal entities (those classified to genus genera level)."***
***But line 236 says "394 benthic foraminifera individuals identified as genera level" the***
***numbers are inconsistent.***

Those were mistakes which have been amended in the reviewed version of the manuscript. Please refer to the new BENFEP_v1 files and Supplement section.

Changes to BENFEP_v1 files:
The species numbers we presented in the submitted version have changed in the new reviewed version because we have received additional datasets (Mallon, 2011; Glock et al., 2020; Tetard et al., 2021) and we have included them. Besides, we have added the living counts of Smith (1964, 1973), Patterson et al. (2000), removed some samples with zero total abundance (wrongly included in the previous submission), and we have detected a few errors which are now amended.

***Line 202: I'm not clear how a sample taken at 0-2 does not contain the surface of***
***sediment and is thus "deeper sampling." Surely a "surface sample" must also have some***
***width to it.***

"Deeper sampling" was changed to "slightly deeper sampling" in the new version of the manuscript in order to clarify this aspect.

***If the original data were given as a proportion rather than a percent, was this converted to***
***a percent and reported as a percent even if not originally given in that form? If so,***
***information about the precision of the originally reported data may not be preserved.***

Belanger et al. (2016) dataset is the only entry provided in proportion. We multiplied their data per 100 to get the percent, and we have now indicated it in "remark_7".

***Figure 5: I don't quite understand the notation in the legend. Would a sample sieved at***
***150 um be colored red or blue? I assume the ( vs ] are telling me the answer, but I'm not***
***familiar with the notation.***

We are sorry that the figure was not clear enough. We now provide an alternative version of the figure. We believe this new plot is clearer than the former one.

Changes to the Figures: new Figure 4 (previously Figure 5).
In the reviewed version of the manuscript, we have re-done Figure 4. We have calculated the relative contribution of each size fraction (>61, >62, etc) independently and represented it as a portion of the pie chart in the new Figure 4.

***Figure 6: The color scaling is a bit odd because all they are not equal width. The blue bar would only***
***be 5-10 species whereas all the others are wider bins. What would this plot look like if it didn't only***
***have valid species but also contained the sp. A, sp. B designations for each genus? This plot may show***
***more about taxonomic resolution of the data than anything real about the diversity of genera. Some***
***discussion about how to appropriately interpret this plot would be warranted.***

In the reviewed version of the manuscript, we now provide an alternative version of the figure. We believe this plot is much clearer than the former one.

Changes to the Figures: new Figure 5 (previously Fig. 6)
In the new version of the manuscript, we provide a new version of Figure 5 with equal spacing in the colour scale and with the revised taxonomic list of BENFEP_v1 (see new Supplement File 1). For this figure, we now consider only the species-level assignations per genus (not taxa below species level).

The same plot but with taxa below species level and designations to genus level ("sps") is provided below.

We consider that showing the number of species-level assignations per genera is more meaningful that counting all the "sps" and varieties which could artificially increase the number of "species".

[Figure]

*Figure 7: Plotting absences in C is a bit odd because an absence at a locality does not necessarily indicate that the taxon did not live there. Blue dots on a blue ocean are very hard to see and the majority of points are some shade of blue or green in this figure.*

We agree that absence does not necessarily mean that the taxon did not live there. We wanted to indicate the presence of a species plus the whole sample set in BENFEP_v1. The latter was wrongly indicated by 0. In the new version of the manuscript, we have replotted Figures 6A-E

Changes to the Figures: new Figure 6 (formerly Figure 7)
In the new version of the manuscript, we provide a new version of the Figure 6C where species "presence" is indicated separately from "station". The colour scales of the other figures 6A, B y D have been also changed to improve the readability.

***In D, it seems that "biodiversity" may be heavily affected by sample sizes and taxonomic decisions and probably should just be called "number of taxa present in database" rather than some measure of "diversity."***

The Reviewer´s comment is right. We acknowledge his/her suggestion and changed the figure caption accordingly to "number of taxa".

---

## Author Comment (AC4)

**Reply to comment on essd-2022-324 (**https://doi.org/10.5194/essd-2022-324-RC3) by Ellen Thomas

The authors would like to thank Ellen Thomas for the time on reading and reviewing the manuscript. We sincerely acknowledge all the Reviewer suggestions that aimed to improve the manuscript readability and usability of the BENFEP database.

Our responses (plain text) to point-by-point comments are listed after Reviewer comments (italic and bold font) followed by the main changes to the manuscript (italic font). Please, refer to the annotated manuscript and companion files.

*Remark 1*

***In my opinion research on paleoceanographic use of benthic foraminifera is indeed (as stated by the authors of this manuscript) greatly hampered by the lack of integration of (the many) published data, which at least in part is caused by the problems on integrating taxonomy. In this manuscript the authors 'unify' the taxonomy to the WORMS database, which is a good first goal. However, the authors do not evaluate the taxonomic concepts of different authors, and do not aim to do so. In fact, that task might be difficult to impossible, because none of the compiled papers provides good descriptions and/or figures of all taxa mentioned. This database does not include descriptions and/or figures of the taxa, thus the user remains uncertain about the uniformity of the species concepts included in this study. I would think that definitely some of the more 'difficult taxa', e.g., various species names of Gyroidina/ Gyroidinoides, of Lagena, of Lenticulina could well be used differently by different authors, a problem not (fully) resolved by the synonymies given in WORMS. One also cannot exclude authors making mistaken identifications. As an example, Nuttallinella florealis White has been extinct since the end Paleocene benthic foraminiferal extinction, thus is unlikely to have been found in surface samples, and was probably misidentified by Liu, 2001. Obesopleurostomella brevis and many other uniserial taxa with a complex aperture became extinct in the mid-Pleistocene benthic foraminiferal extinction event (Hayward et al., 2012, Cushman Foundation Foram Res vol 43), and have been marked as 'fossil only' in WORMS. Taxa with such a relatively recent extinction date have been found in surface samples by e.g., Brady in the Challenger Report: see Hayward and Kawahata, 2005, Extinct foraminifera in Brady's Challenger Report, J Micropalont. 24, 171-175, who list O. brevis as such, as well as P. alternans, also listed in the Supplement. This of course might be caused by the age of the surface-sediment sample not being not well controlled (see below, Remark 2). The authors should in my opinion say something about the possibility of mistaken identification, possibly where they are discussing taxa mentioned by one author only (lines 244-on), or where they are discussing 'taxonomic concepts (section 3.6.1). In my opinion, this lack of detailed checking of the taxonomy definitely does not invalidate the present manuscript. The goal of comparing data from different authors (thus taxonomic 'schools of thought'), as well as different collection methods is a very good and worthwhile first step: after all, the Paleobiology Database is extremely useful for the community without all taxonomic confusion having been solved. In fact, this database effort will make taxonomic evaluation in the future easier to do since so many disparate datasets are here collated. However, I think that the authors should mention the potential problems due to taxonomic misidentification and non-realized synonymies.***

We acknowledge the suggestions made by the Reviewer. This is an interesting question which is not limited to the present study, but to all micropalaeontological databases that include data generated by different authors. Unsatisfactory documentation of taxonomic resolution, divergent taxonomic criteria and/or human errors might lead to some errors in the database (e.g., Siccha and Kucera 2017 referenced in the manuscript).

In the reviewed version of the manuscript, section 3.6.1, we now mention the possibility of some species being misidentified.

Changes to the text: Text rephrased (section 3.6.1).
*Despite the effort to harmonize the taxonomy, it is likely that incorporating data from different authors with diverse taxonomic concepts (e.g., there are 499 species identified by a single author) and potential misidentifications (e.g., see Supplement File 4) could have artificially biased the number of species.*

In order to make clear to the potential users the taxonomic concepts provided by each author, and to bring more transparency to the harmonization procedure, we have now included the following documents in the Supplement section:

Changes to the Supplement: Please, refer to new Supplement section.

-The new Supplement File 2 includes the original authors taxonomic concept for each species harmonized in BENFEP (now BENFEP_v1)

-The new Supplement File 3 includes specific remarks on the harmonization procedure.

-The new Supplement File 4 collates the original authors´ indications about potentially reworked species and comments about some potential misidentifications. We cross referenced this file with the "remark_2" of the database.

**Remark 2:**
**The apparent occurrence of various extinct species in the database: I suggest that the authors mark it in the Supplement (Species list) if the species is given as 'fossil only' in WORMS, in order to warn the reader/user. In such a case, the explanation could be one or more of the following: the species may be misidentified (would be interesting to know whether it was tagged as 'living'), the species may not be extinct (in disagreement with WORMS), or the surface sample may in fact not be recent (or even Holocene) in age. Many of the sampling methods described will not recover 'undisturbed' sediment, i.e., the sediment/water interface will not be recovered in most samplers except multicorers. Could the authors say something about this, either in the text or in the description of sample methods? They mention this in line 296, but could they have added a bit more information, e.g., as described above?**
**It would e.g., be good to know if the authors of the cited article did radiocarbon dating to determine a Holocene age. It would be specifically of interest to know if species described as 'fossil only' in WORMS had been stained and recognized as 'living'.**

We acknowledge the Reviewer suggestion. Indeed, BENFEP_v1 includes 109 taxa identified to species-level considered "fossil only" by WoRMS. The "fossil only" species are treated as the considered extant species. We justify our decision with the following arguments. Firstly, we wanted to provide the users with an unabridged version of the original data. We followed the same criteria for any other non-numerical data originally provided by the authors (section 2.2). Secondly, as the Reviewer indicates in her comments, there are several reasons why a benthic foraminiferal species taken from a surface sediment sample might appear as fossil: a) It is an actual fossil specimen. In that case, the species might be reworked from nearby older sediments; b) it could represent a mistaken identification in the original source or c) the species is not a "fossil only" species but it has been inaccurately attributed as such by WoRMS. The latter is subject to change as new knowledge emerges; therefore, we consider important to retain the information as it appears in WoRMS. Eventually, this study might lead to revisiting the "fossil range" of some species.

In the new version of the manuscript, we have added the following information that could help the users to find whether a species is extinct or extant (according to WoRMS):

-In new section 2.4 we have now indicated that harmonized species include those considered "fossil only" according to WoRMS.

Changes to the text: Text added (new section 2.4, Taxonomic harmonization, previously section 2.5).
*Some taxa included in BENFEP_v1 are considered as "fossil only" by WoRMS. Nevertheless, we kept those in the database. There are several reasons to explain the occurrence of a species categorized as "fossil only" in a sample; it represents a true displaced fossil species from ancient sediments (reworking), a mistaken identification, and, an extant species inaccurately attributed as "fossil" by WoRMS. As it is not clear which of these circumstances applies in each case, we decided to maintain the species to prevent information losses in case of future re-evaluation of the "fossil range" by WoRMS.*

-Changes to the Supplement: Please, refer to the new Supplement section.

The new Supplement File 1 now includes a column dedicated to the "fossil range" according to WoRMS. We re-named the column as "occurrence" to indicate whether a species is extinct or extant. The Supplement File 1 incorporates more comprehensive taxonomic information (status, rank, order, family, class, AphiaID, etc) about the species harmonized in BENFEP (now BENFEP_v1) than the previously submitted file.

-Changes to the BENFEP_v1 files: Please, refer to the new files.
The BENFEP_v1 database will be provided in short and long format. The long format was specifically suggested by three reviewers. Please, refer to the new section 2.5 of the reviewed version of the manuscript (integrates former sections 2.4 and 2.6). There we explain the structure of the two formats of the database.

The long-format version integrates all the information provided in the short format plus taxonomic information for each species (File 1) and each author´ species assignations (File 2). Therefore, users of BENFEP_v1_long will have the information about the taxa "occurrence" in a column alongside the taxonomic designation.

-In new section 3.6.4 we have expanded on the information collated from the publications about age estimates of the surface sediment, foraminiferal preservation and potentially reworked species. The latter was specifically mentioned by some authors. We indicate that in the "remark_2" column of BENFEP_v1 files, which guides the users to look for specific information in the new Supplement File 4. Please, refer to comment above about this file.

Changes to the text: text added (section 3.6.4, The representativeness of the surface sediment assemblages, previously section 3.6.3).

*One of the purported applications of BENFEP_v1 is to provide a quantitative estimate of recent benthic foraminiferal assemblages that could be later used in palaeoenvironmental interpretations (e.g., Fig. 6). The database integrates quantitative data obtained from oceanic regions with different depositional environments, sedimentation rates, carbonate preservations and types of assemblages, collected over different sampling years and using an array of sampling devices that might result in diversion from recent conditions. For example, dead benthic foraminifera obtained from surface sediments might not be representative of the surface if the sampling device fails to recover the sediment-water interface or sedimentation rates are very low. The 36% of the surface sediment samples were retrieved using different types of coring devices (gravity, piston, dart and Phleger corer, calculations using "dev_1"), which are sampling techniques that can cause perturbation or miss-sampling of the surface sediment (Weaver and Schultheiss, 1990). Since the studies included in our database did not date the surface sediment (except for Palmer et al., 2020), we cannot discard that some samples correspond to pre-Holocene conditions. The most comprehensive compilation of sedimentation rates from core-top samples is from the equatorial Pacific and shows highly variable values, ranging from 0.8 to 14.2 cm/ka (Mekik and Anderson, 2018), meaning that surface sediment samples in this region correspond to recent conditions (assuming that no perturbation occurred during sampling). Reworking, downslope transport and carbonate preservation might be other factors influencing the composition of the assemblages obtained from the surface sediments. The presence of "potentially fossil" species reworked from ancient outcrops (see "remark_2" and Supplement File 4) is included in the datasets of Bandy and Arnal (1957), Echols and Armentrout (1980), Ingle et al. (1980) and Zalesny (1959). Still, they represent less than 5% of the assemblage. The contribution of specimens displaced specimens from shallower locations is also low, as indicated by Bandy and Arnal (1957); Ingle et al. (1980); Harman (1964), Pettit et al. (2013), Uchimura et al. (2017) and Zalesny, 1959.*

*Finally, Pettit et al. (2013) in the Gulf of California, and Boltovskoy and Totah (1987) and Resig (1981) off South America in samples below the carbonate compensation depth, are the only authors mentioning poor preservation of calcareous benthic foraminifera.*

***Remark 3:***
***I am not sure whether I am convinced by the authors that the eastern equatorial Pacific should get priority for such a compilation (maybe there are more data for the Atlantic?). However, the question whether this is the highest priority region on Earth for such a compilation - which could be made because we are looking at a large part of the largest ocean on Earth - is unimportant. After all, we must start somewhere with a global database, and exactly where we start does not really matter - as long as the database is open for extension in the future (see below).***

We totally agree with Ellen Thomas on the idea that data sharing and homogenization will definitively enlarge our knowledge about the biogeography of benthic foraminifera, starting from the Eastern Pacific.

The Eastern Pacific was prioritized because authors' previous expertise includes work in this region, and thus being more familiar with the existing literature in this region (e.g., Diz et al 2018, 2020). Besides this "convenience", the OMZ in the Eastern Pacific are the most extensive in the Earth's ocean. Since numerous studies reconstructing the OMZ used benthic foraminifera as the main proxy, we consider that improving the knowledge about the biogeography of this group would be highly beneficial for the palaeocommunity.

In the reviewed version of the introduction, we have indicated the OMZs as a unique feature of the Eastern Pacific and clarified the scientific motivation to build a database for this region.

Changes to the text: Text rephrased (section1, Introduction).
*In an historical context, the increase in ocean temperatures and the expansion of the already existing extensive oxygen minimum zone (found about 100 to 900 m water depth, Karstensen et al., 2000) are the major threats to shallow and deep-water benthic ecosystems from the Eastern Pacific (Sweetman et al., 2017; Breitburg et al., 2018; Yasuhara et al., 2019). These attributes make the Eastern Pacific an area of interest for assessing the past, present, and future of the marine ecosystem status and its response to expected environmental changes (e.g., Calderon-Aguilera et al., 2022).*

***Remark 4: The authors mention that BENFEP is indeed open to such expansion (e.g., lines 23-24), and my earlier remarks indicate that it would indeed be important for the community if BENFEP can be expanded. However, the potential pathway(s) to such extension, e.g., into different oceans, adding additional sources (e.g., articles by Saidova published in Russian), is not very clearly spelled out. How do the authors envisage to keep BENFEP going and growing?***
***Are they proposing to supervise and/or impose quality control on the database? Will there be an opportunity for community input? Which entity (university? Museum? Association?) will; hoist the database in the long term? Potential cooperation which groups hosting databases for other microfossils?***

Unfortunately, we could not have access to the meaningful work of Saidova (1975) for the Pacific Ocean. This is the reason why we did not mention it in the manuscript.

The Reviewer is right in that we did not expand enough on the updating process. In order to facilitate updating, we have re-named BENFEP to BENFEP_v1 along the manuscript and in the PANGAEA repository. Besides, we have now clarified in the reviewed version of the manuscript how we plan to update BENFEP_v1 in the future.

Changes to the text: Text added and rephrased (new section 5. Data availability and future plans, previously section 6).
*The BENFEP_v1 database can be accessed from https://doi.org/10.1594/PANGAEA.947086 (Diz et al., 2022b). This database is conceived as a springboard to store future quantitative data of benthic foraminifera in the East Pacific and make them available to the scientific community. It will be open for any new quantitative data entry and thus, it welcomes any new data published or provided by any contributor. The database will be updated by the authors once a considerable number of new entries need to be incorporated or changes are required to update taxonomic categories to an existing version. New*

*versions of BENFEP will be submitted and curated in PANGAEA. Collaborations with individual researchers and institutions are welcomed specially regarding potential expansion to other ocean basins.*

The current reviewing processes and several interactions during conferences over the last couple of months have been insightful in understanding which the needs and the interest of the benthic foraminifera community are. Expanding BENFEP_v1 and extending the concept to other oceans requires external funding (e.g., for quality control and to host the database in a dedicated server and to migrate and upgrade it to PostGIS or similar), active community input and establishing/strengthening collaborations with other researchers and open access repositories.

We will undertake steps in that direction, and it is in our mind to expand the scheme presented here (i.e., the improvements that might come) to other basins.

*Notes by line number:*
*23-24: the authors say ' We complement BENFEP with an additional database integrating metadata and stations geolocation of benthic foraminiferal studies dearth of quantitative data'. I do not understand the English in this sentence. I see in lines 93-94 and section 4 (lines 305-on) that this is a database containing qualitative record, but this sentence does not explain that. Maybe the authors want to say that they add these because there are so few quantitative datasets? What data does this data What data does this database contain: presence -absence, or such data as 'rare, few, common, abundant'?*

It was unfortunate the way we named the database BENFEPqual and the way we described its content because resulted in confusion.

"BENFEPqual" contained geolocations of samples which carried out benthic foraminifera assemblage studies but whose data were not available in the publications but presented in graphs, as non-quantitative information (rare, dominant, abundant) or species presence. The application of "BENFEPqual" is to geolocate samples/studies in the Eastern Pacific, identify the author, and the type of data and refer to the original source for species information.

Because this dataset is creating confusion, we re-organized the text referring to it as follows:

-We now refer to the dataset as a collection of geolocations of studies that could not be included in BENFEP_v1 because they do not meet the criteria for quantitative data. We provide access to the file in the main text and in the caption of Figure B1.

Changes to the text: Text added (section 2.2).
*There are 31 documents published between 1929 and 2019 characterizing assemblages of living and dead assemblages of benthic foraminifera from surface sediments in the Eastern Pacific that could not be incorporated in BENFEP_v1 because species assemblage data are provided in graphs, as species presence or range of abundances (e.g. common, rare, abundant). The geolocation of the samples and the authors of those publications can be accessed from https://doi.org/10.1594/PANGAEA.947114 (Diz et al., 2022a) and they are represented in Figure B1.*

*Figure B1. Spatial distribution of samples in the Eastern Pacific from studies which do not provide quantitative assemblage data. The numbers refer to each author´s dataset. Sample geolocation and metadata can be found in https://doi.org/10.1594/PANGAEA.947114 (Diz et al., 2022a). The procedure for stations´ georeferencing and column coding follows the indications of sections 2.3 and 2.5. The map was made using ArcGIS software version 10.8.2. The global relief model integrates land topography and ocean bathymetry (Sources: Esri, Garmin, GEBCO, NOAA NGDC, and other contributions).*

-We have removed references to the database by the name of "BENFEPqual" along the text.
Consequently, we have eliminated previous section 4 "Complementary information to BENFEP: BENFEPqual"

We considered that these changes will contribute to alleviate confusion and improve the readability of the text.

*36-37: for the eastern equatorial Pacific, I think that the statement ..'ongoingdeoxygenation..induced by coastal eutrophication' is incorrect or at least incomplete. After all, the eastern equatorial Pacific is the location of the largest open-ocean Oxygen Minimum Zone not linked to coastal eutrophication (see e.g. Breitburg et al., 2018, Declining oxygen in the global ocean and coastal waters. Science 359, eeam7240). Comparison of figure 1 and the figure in Breitburg et al shows clear overlap of the studied sites with that open ocean OMZ, and the authors refer to deep-water deoxygenation in line.*

The Reviewer is right. Therefore, we have rephrased the sentence in this section to make clear that the Eastern Pacific is an open ocean OMZ region, and it is not influenced by ocean eutrophication.

*164-165: The authors say: 'see Supplement for full species description'. Either I am missing something, or the supplement contains a full list of species, but there is no description of species (i.e., description of their morphology). Please change this text to explain what actually is in the Supplement.*

We have re-phrased the sentence referring to Supplement File 1 as follows:
*File 1 indicates the systematics of benthic foraminiferal species listed in BENFEP_v1 following the concepts of the World Foraminifera Database (Hayward et al., 2022, last accessed on 22-12-08).*

*181-190: Text and Caption figure 2. why use words such as 'epipelagic', mesopelagic, bathypelagic, etc.? I think that the words 'xxx-pelagic' are defined for planktonic and nektonic organisms, not for benthos. For example, in general (and in oceanographic textbooks) benthic organisms are described as bathyal or hadal, planktic organisms as bathypelagic or hadal pelagic. The cited paper by Costello & Breyer does not use these xxxpelagic terms for benthos, they use them for planktonic organisms, and for benthics they give the not very useful word 'deep-sea' for benthos below 2000 m. Benthic foram users commonly follow van Morkhoven et al., 1986 depth zones:*
*< 200 m depth: Neritic*
*• coastal: 0–30 m,*
*• inner neritic: 30–50 m,*
*• middle neritic: 50–100 m*
*• outer neritic; 100–200 m*
*200-2000 m depth: bathyal*
*• 200-600: upper bathyal*
*• 600-1000: middle bathyal*
*• 1000-2000: lower bathyal*
*>2000 m depth: abyssal*
*• 2000-3000: upper abyssal*
*• >3000 m: lower abyssal*

Including Costello and Bryden pelagic divisions was a mistake. We have changed the bathymetric ranges to follow the ones indicated by the Reviewer (and described by Morkhoven et al., 1986) and we have modified Figure 2, Figure 6E (former Fig. 7E) and figure captions accordingly. Please, refer to the new figures.

*Figure 2: Distribution of samples with water depth and latitude. Horizontal dashed lines separate the neritic (0- 200 m), the bathyal (200-2000 m) and, the abyssal zones (>2000 m) following bathymetric divisions of van Morkhoven et al. (1986).*

*Figure 6: Geospatial representation of selected species relative abundance (A, B) and presence (C), total number of taxa (D) and selected species mean relative abundance with water depth (E) in BENFEP_v1. Water depth ranges in figure E are as follows: neritic (0-200 m), upper bathyal (200-600 m), middle bathyal (600-1000 m), lower bathyal (1000-2000 m) and abyssal (>2000 m).*

*194: delete 'being'.*

Done as suggested

*213-217: maybe add a remark that nowadays Rose Bengal staining is not thought to be very reliable as indicator of actually living specimens at the time of collection?? e.g. Bernhard, J., 2006, Comparison of two methods to identify live benthic foraminifera: A test between Rose Bengal and CellTracker Green with implications for stable isotope paleoreconstructions, Paleoceanography 21, PA4210 states: 'On average, less than half the Rose Bengal–stained foraminifera were actually living when collected'. In my opinion the authors thus should add some information on the concept of 'stained' vs. 'living.*

We acknowledge the point raised by the Reviewer. We agree with her, and we have amended the text accordingly. We have now explained what living, dead, and living and dead (now renamed as living plus dead) means in the reviewed version of the manuscript (section 3.3). We expanded on the issue of using living or dead assemblages for modern analogue construction or to trace changes in the assemblages over time (new section 3.6.4).

Changes to the text: Text added and rephrased (section 3.3).
*The Rose Bengal staining (Walton, 1952) is the only method used by authors to distinguish dead (non-stained) from living (stained) foraminifera at the time of sampling. Living plus dead refers to an assemblage were living (stained) and dead (non-stained) are counted together in the same sample.*

Changes to the text: Text added and re-phrased (new section 3.6.4, previously section 3.6.3).
*BENFEP_v1 includes information of living, dead, and living plus dead assemblages whose suitability for building recent analogues is under discussion among the scientific community. The use of Rose Bengal as "vital" staining could be controversial because attached bacteria or algae, decaying protoplasm of dead individuals might stain, resembling the staining of the protoplasm of a "true" living individual (see review in Schönfeld, 2012). However, it is still the most widely used method to distinguish "living" (stained) from "dead" (non-stained) foraminifera and it is considered reliable if used cautiously. It might be argued that only living foraminifera should be used to consider baseline studies (Schönfeld, 2012). However, it might also be considered that living assemblages represent a "snapshot" of the foraminifera living at the specific time of sampling and do not hold the time-averaged representativeness of the dead assemblages (Murray, 2000).*

In the main text, we referenced the review of Schönfeld (2012), which detailed the development of the methodology of benthic foraminifera over time and alluded to the meaningful findings of Bernhard et al. (2006).

*222-225: The definition of studied grain sizes is confusing. The text states: For example, 62.5% of the samples were analysed in the >61-74 µm size fraction, 6% in the > 88-106 µm size fraction and 10.5% in the > 125-150 µm size fraction. I think this must mean that the lower boundary of the studied grainsize is in the stated interval, e.g. ' the >61-74 µm size fraction' means that the full range of grainsizes >63 µm was investigated (as listed in Table A1). I think this must be so, because that is what most people do (e.g. Loubere 1994). However, the text appears to read either as if only specimens in between 61 and 74 µm were studied, or as if specimens >71 µm were studied, either of which is incorrect. Similarly, people study either the 125 µm or >150 µm size fraction, but definitely do not limit themselves to specimens between 125 and 150 µm. Then the caption of Figure 5 shows different numbers again, i.e. instead of >63 µm or >61-74 µm the piechart shows (60, 80) (which I think is the same size fraction). Please standardize this to make sure that the reader understands what is listed unequivocally.*

We are sorry that the text and the figure were not clear enough. We now provide an alternative version of the figure and rephrased the text accordingly. We believe this new plot is clearer than the former one.

Changes to the text: Text added and rephrased (section 3.3).
*Most of benthic foraminiferal assemblages were analyzed in the smallest size fraction commonly used in benthic foraminiferal studies. For example, 65.4% of the samples were analyzed using 42, 61, 62, 63 and 74 µm as the lower end of size fraction (e.g. assemblages where studied in the >42 µm, >61µm, >62 µm size fraction, etc.). The 6.1% is using 88 and 105µm and 17.7% the 125, 149, 150, 200, 212 and 500 µm as the lower end of size fraction.*

Changes to the Figures: new Figure 4 (previously Figure 5).
In the reviewed version of the manuscript, we have re-done Figure 4. We have calculated the relative contribution of each size fraction (>61, >62, etc) independently and represented it as a portion of the pie chart in the new Figure 4.

***234: 'valid taxa' - I assume this includes subspecies, varieties and such? could the authors add the number of species? below-species level groups may be included in the species by other researchers, thus making species diversity lower. Again, note that some of the species in this database are listed as extinct, and might have been misidentified. The authors mention how many of the species in their list are 'rare' according to Murray 2013. Could they say a bit more about the total number of species - how many of the more common ones are overlapping with Murray?***

Yes, the numbers in the text wrongly included subspecies and varieties. We have now indicated the species, subspecies, and varieties separately in the text (section 3.4) and following the upgraded version of BENFEP_v1. Please, refer to the new Supplement File 1.

Changes in the text: text rephrased (section 3.4):

*The BENFEP_v1 dataset includes a total of 1091 valid taxa (1073 species, 14 varieties, 4 subspecies) plus two taxa of uncertain status (Serpula lobata and Ammonia avalonensis) corresponding to 335 foraminiferal genera belonging to the classes; Globothalamea (64%), Tubothalamea (11.3%), Nodosariata (19.6%), Monothalamea (4.8%).*

Changes to BENFEP_v1 files:

The species numbers we presented in the submitted version have changed in the new version because we have received additional datasets (Mallon, 2011; Glock et al., 2020; Tetard et al., 2021) and we have included them. Besides, we have added the living counts of Smith (1964, 1973), Patterson et al. (2000), removed some samples with zero total abundance (wrongly included in the previous submission), and we have detected a few errors which are now amended.

In the new version of the manuscript, we have briefly discussed some species numbers:

-section 3.4 (Benthic foraminiferal species): *The BENFEP_v1 database contains 292 valid species (excluding varieties and subspecies) that can be considered rare, with a mean relative contribution lower than 1% (Murray, 2013)*
and
-section 3.6.1 (Taxonomic concepts): *Despite the effort to harmonize the taxonomy, it is likely that incorporating data from different authors with diverse taxonomic concepts (e.g., there are 499 species identified by a single author) and potential misidentifications (e.g., see Supplement File 4) could have artificially biased the number of species.*

The number of rare species in BENFEP_v1 is high, much higher than the 120 species for which Murray (2013) provided a detailed biogeography. In despite of the fact that we agree with the Reviewer about the interest of analyzing the species overlapping with Murray, or even with other works, this kind of analysis is far for being simple and it would deserve a detailed study and analysis. In this sense, we believe that due to its complexity, it goes beyond the scope of this study.

***235: please provide reference for these Classes- who defined these?***

These are the classes as they appear in WoRMS. We provide Hayward et al. (2022) as a reference in the figure caption.

*Figure 5: Number of valid species per foraminifera genus and its distribution among the classes indicated by WoRMS (Hayward et al., 2022, last accessed on 22-12-08). Only genera with 5 or more species are represented in the figure.*

*248: are these all (90) at the species level or including below species level taxa?*

We have rephrased the text in the new version of the manuscript: *Furthermore, the highest number of taxa (90) is found in a station studying dead individuals located in the South Pacific at 1800 water depth (Ingle et al., 1980, Fig. 6D).*

The number is from a sample studied by Ingle et al., (1980). The sample does not include taxa below species level but it includes all authors species-level identifications plus "sps".

The number of varieties (14), subspecies (4) in BENFEP_v1 is low compared to the number of total valid taxonomic assignations: 1093 (1091 plus 2 taxa of uncertain status)

**251: typo in Nuttallides (spelled with two letters t)**

This has been changed in the species list.

*Figure 7: figurer 7D says 'Biodiversity' but plots number of species. the Number of species is NOT how diversity is defined, it is 'species richness'.*

The Reviewer's comment is right. We acknowledge his/her suggestion and changed the figure caption accordingly to "number of taxa".

*256: what is a 'heatmap'?*

We have considered that a heatmap (or a density grid) is the best option to visualize the volume of locations or events in a dataset directing the viewers towards areas with intensive survey sampling. It is used to analyse point data by transforming the points into a regular grid. Each resulting grid cell is assigned a value that is determined by the proximity of nearby points. The colour palettes represent the values from low to high.

*261: 'made' or 'drawn' rather than 'elaborated'.*

Amended and changed to "The slides for the video were made using QGIS"

*286: rounding could lead to both >100 as well as < 100%?*

Yes, the Reviewer is right, and rounding might lead to any of both results.

The percentages summing more and less than 100% were a source of continuous concern during the building up of the database, because unfortunately, that is a common feature in data archived as percentages. For datasets that had to be machined or manually digitized we cross-validated the typed entries before adding them to BENFEP_v1. Human errors, either in the original publication or during the digitization step, cannot be completely ruled out.

*316-on (section 5): here the authors should have mentioned that presently many journals require making data available, and will not publish papers without such data, which must be provided not just in the journal, but in an accepted database such as Pangaea. They are escribing some of the broadly accepted FAIR data practices: Findable, Accessible, Interoperable, and Reusable. Their text here should have pointed this out: in many cases 'authors should not be encouraged' - they will be required to do so by reputable journals. -*
*https://www.nature.com/articles/sdata201618*

We agree with the Reviewer that it should be bolder and we rephrased the text to: *Publishers should commit to FAIR data practices (Wilkinson et al., 2016) and the authors must share their published data in a readily accessible format and in public repositories to avoid the irreversible loss of valuable quantitative data.*

It is worth mentioning that some of the datasets included in this database published during the last five years were not available in Pangaea or similar repositories. They had to be machine digitized or requested to their authors. For some others, we obtained negative answers after data were requested and consequently, data could not be incorporated in BENFEP_v1. We also found that some journals that in principle comply with FAIR data practices, published papers which included statements about the data availability such as 'data available upon request', which obviously does not meet the FAIR practices. It would be invidious to cite them here, but these situations just highlight the problems of open-access data still in XXI century.

***Table A1: for studies describing 'living + dead', it is possibly to provide the information how many of these species were observed as 'living', how many as 'dead' (with overlap, some species could be present both living and dead)? it would be good information to know whether species were present both as living and as dead.***

Living plus dead (LD) refers only to studies that specifically stated that they studied living (Rose Bengal stained) and dead (un-stained) assemblages from the same sample. We have now explained what living, dead and living plus dead means in the reviewed version of the manuscript section 3.3, as indicated in the reply to a comment above.

We would like to explain that we did not artificially sum live and dead counts and when we curated "LD" is because the authors provided the living and dead counts combined. However, in three entries we did not include the Living assemblage which the author provided separately. This happened in Smith (1964, 1973) and Patterson et al., (2000). We have now included these living samples in the reviewed version of BENFEP_v1.

***Table B1: I think there are mistakes in N200, N300, which are both described as ' It indicates whether sample counts are equal to or higher than 100 individuals'. I think this should be 200 and 300, respectively.***

The Reviewer is right, they are mistakes. They have been amended and corrected in the reviewed version of the manuscript and Tables C1 and C2 of the Appendices section.

***Note to supplement:***
***There are two entries for Oridorsalis tener (681, 682). meant or mistake?***

The URLs referring to these two entries were different. The mistake appeared because we accidentally removed the variety in one of the species name. This is amended in the new Supplement (File 1), which provides an updated and more comprehensive taxonomic information of the species curated in BENFEP_v1.

---

## Author Comment (AC5)

**Reply to comment on essd-2022-324** (https://doi.org/10.5194/essd-2022-324-RC4) by an Anonymous Reviewer

The authors would like to thank the Anonymous Reviewer for the time on reading and reviewing the manuscript.

Our responses (plain text) to point-by-point comments are listed after Reviewer's comments (italic and bold font) followed by the main changes to the manuscript (italic font). Please, refer to the annotated manuscript and companion files.

Please, find the access to the new database files via a key provided by the editorial office.

*Points of revision for the paper:*

*Line 121-122: "The whole database can be managed using R version 4.2.1 (R Core Team, 2022). It can be uploaded and managed with geographic information system software such as QGIS and ArcGIS after changing the table format from wide to long."*
*It would be great, if the database could be managed with geographic information system software such as ArcGIS. If I would have access to the database, I would have tested this myself. I think it would be really helpful, if the authors provide some more information on how to do this. For example, either the database could be uploaded in different file formats. At the moment the file that is uploaded at Pangaea seems to be an .xlsx file. This is of course great for standard users of .xls based software but makes it more complicated for other users (R for example), especially with such a big file. One solution could be to upload the database in different file versions. One text file in long format that could be directly imported into QGis or ArcGIS, another one in wide format for other applications and the original .xslx file. This would be very helpful for users but also would make updating the database in the future very tedious, since all the files would have to be exchanged. Another possibility would be to publish an R friendly text file that could easily be imported into .xls based software. If this file is in a wide format, it would be easy to provide an R script to convert it into a QGIS/ArcGIS friendly long format, either in the paper or in the supplement. It is also not directly clear to me, which columns would have to be merged to make the format GIS friendly. Here is an example for such a cookbook (using the gather() argument from the tidyr library): (….). These are of course only optional suggestions on how the accessibility of the manuscript might be improved. My other point is not really a change in the manuscript itself but more a question/suggestion about the future of this database.*

Thanks for the Reviewer's suggestions and comments. Because it is our goal to reach all types of potential users interested in benthic foraminifera, regardless of their preferred method to handle large datasets; we have followed the Reviewer's suggestion and provided BENFEP (now BENFEP_v1) in two formats: short and long. Both versions are given in text format. For those who use more conventional software to read spreadsheet-type files, we keep the previously submitted format (BENFEP_v1_short). For those who prefer to access data using other types of software (e.g., geospatial software), we provide the database in long format (BENFEP_v1_long). The long format has been also suggested by Lukas Jonkers and another Anonymous Reviewer. Additionally, we provide general indications on how to manage the short version in R.

We have also implemented changes to the Supplement section to make clear to the potential users the taxonomic concepts provided by each author and to bring more transparency to the harmonization procedure. Some of the information included in the new Supplement files is incorporated in BENFEP_v1_long. Those changes are explained below.

Changes to the Supplement: Please, refer to the new Supplement section.

-The new Supplement File 1 incorporates more comprehensive taxonomic information (status, rank, order, family, class, AphiaID, etc) about the species harmonized in BENFEP (now BENFEP_v1) than the previously submitted file.

-The new Supplement File 2 includes the original authors' taxonomic concepts for each species harmonized in BENFEP_v1 and listed in new Supplement File 1.

-The new Supplement File 3 includes specific remarks on the harmonization procedure.

-The new Supplement File 5 provides general information on how to perform basic operations with BENFEP_v1_short in R

Changes to BENFEP (now BENFEP_v1): long and short formats.

The potential users of BENFEP_v1_short will have the original authors' specific assignations in a separate file (Supplement File 2). Conversely, the long-format version integrates all the information provided in the short format plus taxonomic information for each species (File 1) and each author´s species assignations (File 2).

Please, refer to new section 2.5 of the revised version of the manuscript (which integrates former sections 2.4 and 2.6). There we explain the structure of the two formats of the database. Please, see the new Table C1 and Table C2, where we explain column names and column codes for BENFEP_v1_short and BENFEP_v1_long.

We would like to indicate that BENFEP_v1_short and BENFEP_v1_long can be opened and managed in GIS software, such as QGIS and ArcMap, among others. Both files can be opened as tables and plotted using the coordinate columns (long, lat) and the EPSG:4326 (WGS84). At this point, the user can transform the tables into a GIS format to facilitate the analysis and visualization. In the specific case of the short format database, the users might consider the possibility of managing it as a geodatabase or a geopackage file, because the number of fields is higher than 255 (limit of other GIS formats like shapefiles). In the case of the long format, there are no limitations in this regard.

***Line 338-342: "This database is conceived as a springboard to store future quantitative data of benthic foraminifera in the East Pacific and make them available to the scientific community. It can be enlarged with new records as they are being generated or after the authors request, therefore providing an ongoing live resource. Any changes to add, correct, or update taxonomic categories to an existing version will be indicated in PANGAEA."***
***How do the authors think about the future of the database? Should there be an easy protocol or "cookbook", how to add data of new records as they are being generated? Are there any permanent members of the group that are able to sustain, clean up and update the database? I think for individual members this might be an impossible task but it should be important to avoid misuse of the feature to update the database. One suggestion would be to provide a review system like in WORMS: Uploaded datasets that are not reviewed, yet, will be marked as "not reviewed". Reviewed datasets might be marked as "accepted" or "unaccepted", if there are any issues considered by a reviewer. Reviewers could be volunteer foraminifera taxonomists. I think there might be a big support in the community for such an effort.***

Unfortunately, we do not have at the moment the resources to migrate BENFEP_v1 to a system that enables the reception of external datasets, like WoRMS. We anticipate that the new entries to the database will be communicated via email to the corresponding author or any other of the current team members. After the reception of the dataset, the procedure would be very similar to the one described in the reviewed version of the manuscript, with entails taxonomic harmonization (2.4), curating metadata (section 2.6), and establishing a quality control on the data entered. In the reviewed version of the manuscript, we have now expanded on the taxonomic harmonization procedure.

Changes to the text: Text added (new section 2.4, Taxonomic harmonization, previously section 2.5).
*In order to find the valid species name, we searched each author´s original species assignment in the WoRMS research engine. This procedure enables to identify whether the original species name is accepted (valid species) or if it is a synonymous of the valid species or taxa correspond to a variety or a subspecies. When the original species name was not currently in use, it was substituted by the valid species, subspecies, or variety name.*

***Finally, are there any plans to include other ocean basins into the database in the future? These are just some questions/suggestions about the future of the database and of course, this is an own project by itself and some things cannot be directly integrated into the paper and database. For example, the review system for future datasets would need an own platform or deeper collaboration with Pangaea. Though, I think it is worth it to think about the legacy of such a huge project and maybe to integrate some points about the discussion of the future of the database at the end of the paper about "Data availability and future plans".***

The reviewer is right, and we regret not having delved further into the process of updating the database. In order to facilitate updating, we have re-named BENFEP to BENFEP_v1 along the manuscript and in the PANGAEA repository. Besides, we have now clarified in the reviewed version of the manuscript how we plan to update BENFEP_v1 in the future.

Changes to the text: Text added and rephrased (new section 5. Data availability and future plans, previously section 6).
*The BENFEP_v1 database can be accessed from https://doi.org/10.1594/PANGAEA.947086 (Diz et al., 2022b). This database is conceived as a springboard to store future quantitative data of benthic foraminifera in the East Pacific and make them available to the scientific community. It will be open for any new quantitative data entry and thus, it welcomes any new data published or provided by any contributor. The database will be updated by the authors once a considerable number of new entries need to be incorporated or changes are required to update taxonomic categories to an existing version. New versions of BENFEP will be submitted and curated in PANGAEA. Collaborations with individual researchers and institutions are welcomed specially regarding potential expansion to other ocean basins.*

The current reviewing processes and several interactions during conferences over the last couple of months have been insightful in understanding which the needs and the interest of the benthic foraminifera community are. Expanding BENFEP_v1 and extending the concept to other oceans requires external funding (e.g., for quality control and to host the database in a dedicated server and to migrate and upgrade it to PostGIS or similar), active community input and establishing/strengthening collaborations with other researchers, and open access repositories.

We will undertake steps in that direction, and it is in our mind to expand the scheme presented here (i.e., the improvements that might come) to other basins.